# High-intensity training enhances executive function in children in a randomized, placebo-controlled trial

**David Moreau\*, Ian J Kirk, Karen E Waldie**

Centre for Brain Research, The University of Auckland, Auckland, New Zealand

## Abstract

**Background:** Exercise-induced cognitive improvements have traditionally been observed following aerobic exercise interventions; that is, sustained sessions of moderate intensity. Here, we tested the effect of a 6 week high-intensity training (HIT) regimen on measures of cognitive control and working memory in a multicenter, randomized (1:1 allocation), placebo-controlled trial.

**Methods:** 318 children aged 7-13 years were randomly assigned to a HIT or an active control group matched for enjoyment and motivation. In the primary analysis, we compared improvements on six cognitive tasks representing two cognitive constructs (N = 305). Secondary outcomes included genetic data and physiological measurements.

**Results:** The 6-week HIT regimen resulted in improvements on measures of cognitive control [BFM = 3.38, g = 0.31 (0.09, 0.54)] and working memory [BFM = 5233.68, g = 0.54 (0.31, 0.77)], moderated by BDNF genotype, with met66 carriers showing larger gains post-exercise than val66 homozygotes.

**Conclusions:** This study suggests a promising alternative to enhance cognition, via short and potent exercise regimens.

**Funding:** Funded by Centre for Brain Research.

**Clinical trial number:** NCT03255499.

**\*For correspondence:**
d.moreau@auckland.ac.nz

**Competing interests:** The authors declare that no competing interests exist.

## Introduction

Possibly the most reliable means to induce cognitive improvements behaviorally, physical exercise has become known for its effects on the brain in addition to its well-documented impact on the body (see for a review *Moreau and Conway, 2013*). Individuals with higher cardiovascular fitness typically show higher performance on a wide range of cognitive measures, from cognitive control (*Pontifex et al., 2011*) to working memory (*Erickson et al., 2013*) and executive functioning (*Colcombe and Kramer, 2003*; *Hillman et al., 2008*). In addition, long-term sport practice is associated with higher working memory capacity (*Moreau, 2013*), spatial ability (*Moreau, 2012*), and more efficient visual processing of movements (*Güldenpenning et al., 2011*). In neuroimaging studies, greater fitness indices have also been linked to differences in white matter integrity (*Chaddock-Heyman et al., 2014*; *Voss et al., 2013*), as well as with larger hippocampal (*Weinstein et al., 2012*) and cortical areas (*Erickson et al., 2009*; *Makizako et al., 2015*). These results are corroborated by increased long-term potentiation (LTP) in the visual system of physically fit individuals compared to the non-fit (*Smallwood et al., 2015*), a key finding given the primary role of LTP in major cognitive processes such as learning and memory (*Bliss and Collingridge, 1993*, *Bliss and Lomo, 1973*).

Importantly, these correlational findings are further supported by longitudinal designs. Exercise interventions can elicit cognitive improvements in diverse populations ranging from children (*Davis et al., 2011*) to the elderly (*Fabre et al., 2002*), as well as in individuals with various clinical conditions such as developmental coordination disorders (*Tsai et al., 2012*) and schizophrenia

**eLife digest** Exercise has beneficial effects on the body and brain. People who perform well on tests of cardiovascular fitness also do well on tests of learning, memory and other cognitive skills. So far, studies have suggested that moderate intensity aerobic exercise that lasts for 30 to 40 minutes produces the greatest improvements in these brain abilities.

Recently, short high-intensity workouts that combine cardiovascular exercise and strength training have become popular. Studies have shown that these brief bouts of strenuous exercise improve physical health, but do these benefits extend to the brain? It would also be helpful to know if the effect that exercise has on the brain depends on an individual's genetic makeup or physical health. This might help to match people to the type of exercise that will work best for them.

Now, Moreau et al. show that just 10 minutes of high-intensity exercise a day over six weeks can boost the cognitive abilities of children. In the experiments, over 300 children between 7 and 13 years of age were randomly assigned to one of two groups: one that performed the high-intensity exercises, or a 'control' group that took part in less active activities – such as quizzes and playing computer games – over the same time period. The children who took part in the high-intensity training showed greater improvements in cognitive skills than the children in the control group. Specifically, the high-intensity exercise boosted working memory and left the children better able to focus on specific tasks, two skills that are important for academic success.

Moreau et al. further found that the high-intensity exercises had the most benefit for the children who needed it most – those with poor cardiovascular health and those with gene variants that are linked to poorer cognitive skills. This suggests that genetic differences do alter the effects of exercise on the brain, but also shows that targeted exercise programs can offer everyone a chance to thrive.

Moreau et al. suggest that exercise need not be time consuming to boost brain health; the key is to pack more intense exercise in shorter time periods. Further work could build on these findings to produce effective exercise routines that could ultimately form part of school curriculums, as well as proving useful to anyone who wishes to improve their cognitive skills.

(*Pajonk et al., 2010*). Furthermore, improvements appear to be dose-dependent (*Davis et al., 2007*; *Vidoni et al., 2015*), and are not restricted to atypical or clinical populations—adults at their cognitive peak show similar benefits (*Gomez-Pinilla and Hillman, 2013*; *Moreau and Conway, 2013*; *Voss et al., 2011*).

In school settings, exercise interventions have shown to be associated with higher levels of academic achievement (*Coe et al., 2006*), and exercise regimens in children typically lead to improvements in various aspects of cognition, including executive function, cognitive control and memory (see for a review *Tomporowski et al., 2015a*). Interventions implemented in early stages of life allow capitalizing on higher cortical plasticity, potentially maximizing their impact (*Cotman and Berchtold, 2002*). The appeal of early interventions has motivated a whole line of research exploring the effect of physical exercise regimens on behavior, cognitive function, and scholastic performance (*Castelli et al., 2007*; *Davis et al., 2007*, *Davis et al., 2011*; *Donnelly et al., 2016*; *Jackson et al., 2016*; *Pontifex et al., 2013*). Consistent with these findings, Sibley et al. found a robust effect of physical exercise on cognitive function in children, in a meta-analytic assessment of the literature at the time (*Sibley and Etnier, 2003*).

Most studies in this line of research have evaluated the impact of a rather specific type of regimen, based on aerobic exercise. Usually defined as a sustained regimen performed at moderate intensity (e.g., *McArdle et al., 2006*), aerobic exercise has come to be accepted as the form of exercise typically associated with neural changes and cognitive enhancement (*Hillman et al., 2008*; *Thomas et al., 2012*), for at least two reasons. First, current interventions are rooted in early findings in the animal literature, which typically investigated the effects of physical exercise in rodents—animals who naturally favor aerobic forms of exercise (*Gould et al., 1999*; *Shors et al., 2001*). Second, the most dramatic gains in cognition have been observed in the elderly (*Erickson et al., 2015*; but see also *Etnier et al., 2006*; *Young et al., 2015*, for a more nuanced view), a population for which

moderate-intensity exercise is seemingly most adequate. Subsequent studies have stemmed from this line of research, therefore expanding the initial paradigm to a wider range of populations.

Yet current trends of research suggest other promising directions. For example, regimens based on resistance training have shown sizeable effects on cognition (*Best et al., 2015*; *Liu-Ambrose et al., 2012*; *van Uffelen et al., 2007*), despite underlying mechanisms of improvements being potentially different from those elicited by aerobic exercise (*Goekint et al., 2010*). More complex forms of motor training that combine high physical and cognitive demands also appear encouraging (*Moreau et al., 2015*; *Tomporowski et al., 2015b*). Moreover, a compelling body of research in the field of exercise physiology indicates that interventions based on short, intense bursts of exercise can induce physiological changes that mirror those following aerobic exercise on a wide variety of outcomes. These include measures of cardiovascular function (*Chrysohoou et al., 2015*; *Gayda et al., 2016*), overall fitness (*Benda et al., 2015*), and general health (*Milanović et al., 2015*). In some cases, physiological improvements following high-intensity training (HIT) can even go beyond those typically following aerobic regimens (*Rognmo et al., 2004*). For example, HIT appears to be particularly effective to increase the release of neurotrophic factors essential to neuronal transmission, modulation and plasticity—potentially surpassing aerobic exercise regimens (*Ferris et al., 2007*). This body of research is promising, as it suggests a plausible mechanism by which intense bursts of exercise could meaningfully influence cognitive function and behavior.

A few studies have partially tested this idea. Short bouts of exercise have been shown to alleviate some of the difficulties typically associated with Attention-deficit/hyperactivity disorder (ADHD) in children (*Piepmeier et al., 2015*), demonstrating the potential of this type of intervention to enhance cognitive abilities. The benefits reported in this study were not limited to children diagnosed with ADHD, however—typical children also exhibited cognitive improvements. More strikingly perhaps, Pontifex and colleagues found that a single 20 min bout of exercise was sufficient to improve cognitive function and scholastic performance in children (*Pontifex et al., 2013*). This is an impressive finding, given that exercise-induced cognitive improvements typically occur after longer training periods (*Etnier et al., 2006*; *Sibley and Etnier, 2003*). Importantly, these effects should be distinguished from short-term improvements immediately following acute bouts of exercise (*Tomporowski, 2003*), which typically dissipate after a few hours. The two types of outcomes (short-term consequences of single acute sessions vs. more durable benefits) are sometimes conflated, resulting in misleading conclusions (*Jackson et al., 2016*). The hypothesized mechanisms are, however, different—heightened state of alertness induced by neurotransmitter increases for the former (*Tomporowski, 2003*), and slower but more durable neurophysiological adaptations in the case of the latter (*Erickson et al., 2011*, *Erickson et al., 2015*; *Moreau and Conway, 2013*; *Voss et al., 2013*).

One aspect that remains to be formerly investigated relates to the specific influence of exercise intensity. Although focused on short sessions, the aforementioned studies utilized regimens of moderate intensity—a reported 62–72% (*Piepmeier et al., 2015*) and 65–75% (*Pontifex et al., 2013*) of individual maximum heart rate, respectively. Yet based on findings from the physiological literature (e.g., *Rognmo et al., 2004*), there are clear mechanisms via which exercising at a high intensity could influence cognition in a meaningful manner. Arguably, HIT could elicit improvements above and beyond those typically following short sessions of moderate intensity, and provide a legitimate, time-efficient alternative to longer aerobic exercise regimens (*Costigan et al., 2015*). Together, the conjunction of promising early findings and clear mechanisms of action has prompted discussions to implement HIT interventions more systematically within the community (*Gray et al., 2016*).

In an effort to better understand and predict individual responses to physical exercise interventions, several studies have investigated the role of specific genetic polymorphisms on the magnitude of exercise-induced improvements (*Erickson et al., 2008*, *Erickson et al., 2013*). Among these, many have focused on the brain-derived neurotrophic factor (*BDNF*) val[66]met polymorphism, given its direct influence on serum *BDNF* concentration (*Lang et al., 2009*). *BDNF* is known to support neuronal growth and has been shown to facilitate learning, a process that in turn induces *BDNF* production (*Berchtold et al., 2001*; *Cotman and Berchtold, 2002*; *Kesslak et al., 1998*). This dynamic coupling makes *BDNF* an important underlying factor of exercise-induced cognitive improvements. Consistent with this idea, it has been proposed that individuals whose particular *BDNF* polymorphism is associated with lower activity-dependent *BDNF* levels (met[66] carriers) might benefit from exercise interventions to a greater extent than individuals whose activity-dependent *BDNF* levels are higher (val[66] homozygotes).

Similarly, a few studies have shown that individuals with lower cardiovascular function might maximize benefits from physical exercise interventions designed to improve cognitive function (*Sofi et al., 2011*; *Strong et al., 2005*). The implicit assumption is that although lack of physical exercise might be a limiting factor for individuals whose fitness level is low, more active individuals might be less impacted by an exercise intervention program (*Heyn et al., 2004*; *Lautenschlager et al., 2008*; *Sniehotta et al., 2006*). Despite the intuitive appeal of this assumption, several studies, including one from our group, have failed to find a positive correlation between exercise-induced cognitive improvements and the associated physiological changes (*Moreau et al., 2015*; *Tsai et al., 2014*). Arguably, the absence of a clear link between the hypothesized physiological mechanisms of improvements and tangible cognitive gains might stem from the plurality and complexity of variables underlying these changes, rendering coherent associations elusive.

Despite similarities in the physiological mechanisms linking aerobic exercise and HIT on cognition, the precise impact of the latter on cognitive performance remains to be confirmed experimentally. In the present study, we tested the viability of HIT as a substitute for aerobic exercise to induce cognitive improvements in school populations. In particular, we postulated that HIT would result in improvements in measures of cognitive control and working memory, as both constructs have been linked to fitness levels (*Pontifex et al., 2011*) and appear to be malleable via aerobic regimens (*Erickson et al., 2013*). The choice of these constructs was also motivated by previous research showing the malleability of both cognitive control and working memory in training studies, thus providing theoretical and empirical support for the plausibility of expected improvements (*Hampshire et al., 2012*; *Mishra et al., 2014*). Consistent with recent efforts to better model and understand the mechanisms of cognitive improvement (*Young et al., 2015*; *Moreau and Waldie, 2015*), the present study also intended to address interindividual variability so as to isolate the underlying factors of improvement. Based on previous literature (*Erickson et al., 2013*; *Moreau et al., 2015*), we hypothesized that exercise training would elicit substantially larger cognitive benefits in individuals whose cardiovascular fitness is low, and in *BDNF* met[66] carriers, whose activity-dependent *BDNF* levels are naturally limited. Finally, we expected physiological improvements with exercise, as typically induced from aerobic interventions (see for a review *Gomez-Pinilla and Hillman, 2013*).

## Results

Statistical analyses were performed in R (RRID:SCR_001905; *Core Team R, 2016*). The following R packages were used for our analyses (in alphabetical order): BayesFactor (*Morey and Rouder, 2015*), car (*Fox and Weisberg, 2011*), dplyr (*Wickham, 2011*), ggplot2 (*Wickham, 2009*), gridExtra (*Auguie, 2012*), lsr (*Navarro, 2015*), psych (*Revelle, 2015*), pwr (*Champely, 2015*), rjags (*Plummer, 2016*). All packages were retrieved from CRAN (RRID:SCR_003005; https://cran.r-project.org/). Figures 4–6 were generated in JASP (*JASP Team, 2016*). R code and data are freely available on GitHub (https://github.com/davidmoreau/2017_eLife; a copy is archived at https://github.com/elifesciences-publications/2017_eLife-1). The repository includes data sets, R scripts, details and script of the HIT workout, the CONSORT flow diagram and the CONSORT checklist.

In this section, we report Bayesian model comparisons, to allow quantifying the degree of evidence for a given model compared to other models tested, as well as Bayesian parameter estimations when relevant. All the equivalent frequentist analyses can be found at the end of the Results section.

Normality of distribution was examined for all continuous variables. If distributions were skewed, we compared results using non-corrected vs. log-transformed data, and looked for discrepancies. Although the analyses we present below are fairly robust to outliers, as priors can be adapted to reflect deviations from normality, we systematically checked consistency using standard approaches to outlier exclusion, to facilitate direct comparisons with frequentist tests. We defined outliers as values more than 3/2 times the upper quartile or less than 3/2 times the lower quartile of a given distribution, and systematically checked consistency of our results with and without inclusion.

### Physiological improvements

Participants in the exercise group saw a greater decrease in resting heart rate than controls, as demonstrated by a Bayesian ANCOVA with *Condition* (HIT vs. Control) as a fixed factor and baseline

heart rate as a covariate. The full model was preferred to the model with baseline resting heart rate only: $BF_M = 40.45$, and was the most likely given our data: $P(M \mid Data)=0.93$, assessed from equal prior probabilities (*Figure 1A*). In-depth analyses focused on individuals with elevated resting heart rate at baseline allowed further insights into the potency of our exercise intervention. Specifically, a Bayesian t-test on resting heart rate change showed a sizeable difference between the two groups, with larger gains for the HIT group ($BF_{10} = 3.47$, with $M_{gain} = 6.11$, $SD_{gain} = 11.64$ and $M_{gain} = 1.89$, $SD_{gain} = 8.63$, for HIT and Control, respectively; Hedges' $g = 0.41$ (0.09, 0.73). Test-retest reliability—assessed via a comparison between pretest and posttest resting heart rate for controls only—was acceptable ($r = 0.77$, $BF_{10} = 1.75$ e+51).

Physiological data also provided important indications about workout intensity idiosyncrasies. We used resting heart rate at pretest to determine target intensities for each individual, such that:

$$HR_{Target} = HR_{Reset} + \delta(HR_{Max} - HR_{Rest}) \qquad (1)$$

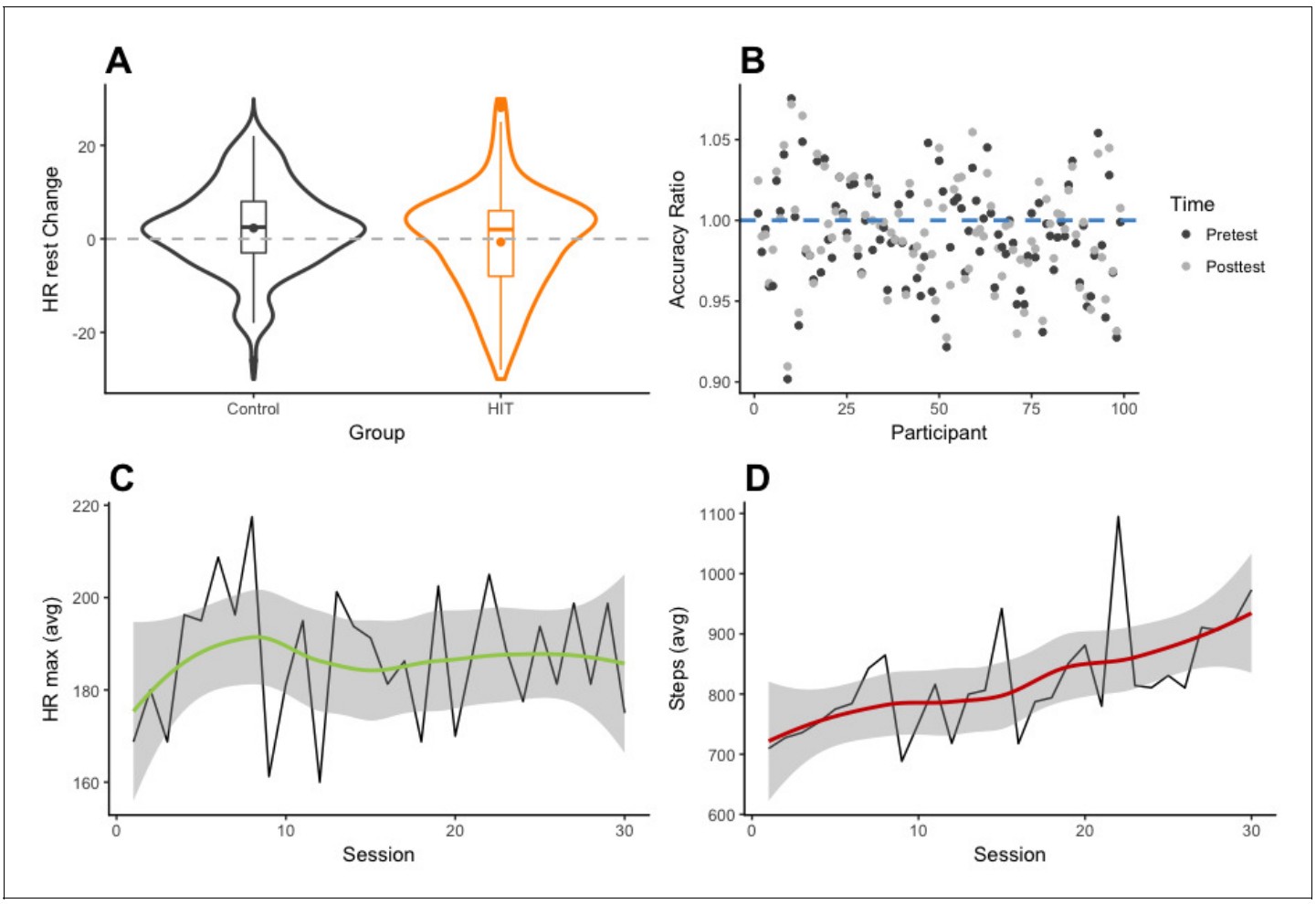

**Figure 1.** Physiological and effort-dependent measures. (A) Violin and box plots showing change in resting heart rate (in BPM) between pretest and posttest sessions, for HIT and control groups. The dashed line shows the point of perfect equivalence between pretest and posttest measurements; values below the line indicate heart rate decreases. (B) Targeted range accuracy, defined as the ratio of maximum measured heart rate per participant (in BPM) to targeted heart rate (expected), averaged across sessions. Dark dots show accuracy based on pretest resting heart rate, whereas light dots show accuracy based on posttest resting heart rate. The blue dashed line represents the point of perfect agreement between individual targeted heart rate and maximum measured heart rate. Values above the line represent higher measured heart rate than expected from baseline. (C) Time series of the maximum heart rate (in BPM) measured for a single workout, averaged over participants, plotted across sessions. Smoothing is modeled via a non-parametric locally weighted regression using a nearest neighbor approach (i.e. local polynomial regression fitting). (D) Time series of the total number of steps for a single workout, averaged over participants, shown across sessions. Smoothing is modeled via a non-parametric locally weighted regression using a nearest neighbor approach (i.e. local polynomial regression fitting).

where $HR_{Max}$ = 220 Age, and $\delta$ is set to. 80. This yielded an individual target range ($HR_{Target}$ or above) while exercising. We then compared this range with the maximum intensity measured during each workout, to obtain an index of accuracy, or agreement, between target zone and actual effort. Results showed that participants did exercise at a suitable intensity overall, as expressed by the deviation from individual target heart rate values ($M_{Dev}$ = 1.65, $SD_{Dev}$ = 5.87; *Figure 1B*). Importantly, effort intensity was maintained stable across time, as demonstrated by moderate evidence favoring the null model over an alternate model that included time as a predictor of maximum heart rate in a Bayesian linear regression analysis [$BF_M$ = 2.87, $P$(M | Data)=0.74, *Figure 1C*]. Because individual resting heart rates tended to decrease throughout the intervention, sustained effort indicates that individuals incrementally increased workout volume, which was confirmed by additional measures such as step count [$BF_M$ = 2.979e + 10, $P$(M | Data) $\approx$ 1, for the model that included *Session* as a covariate, *Figure 1D*]. Together, these results support the notion that the intervention was adaptive, allowing workout intensities tailored to each individual.

Physiological improvements are informative in two key aspects: they provide corroborating evidence for the hypothesized changes associated with exercise, and they allow identifying idiosyncratic parameters often characteristic of training interventions. However, the main goal of a cognitive intervention is to elicit *cognitive* gains, which were the primary outcomes of the present intervention. In the following sections, we first identify latent constructs from cognitive assessments, before discussing the impact of the intervention on these two constructs.

## Exploratory factor analysis

An exploratory factor analysis using principal component extraction and promax rotation was performed on all six cognitive measures at pretest. Although less common than orthogonal rotations, oblique rotations such as promax allow factors to correlate; this property is especially appropriate when the factors extracted are assumed to be correlated to some degree—a reasonable assumption given our design. The corresponding scree plot and eigenvalues (i.e. the variance in all variables accounted for by each factor) suggested a two-component solution (see factor loadings in *Table 1* and *Table 1—source data 1* and *2*). Subsequent test of the two-factor model confirmed that the number of factors was sufficient ($\chi^2$ (4)=0.59, p=0.96; Bayesian Information Criterion, BIC = −22.05). We refer to these two components hereafter as Cognitive Control and Working Memory. The correlation between the two factors was $r$ = 0.32. Uniqueness values indicated that the tasks spanned an adequate range within the sample space of each construct (*Table 1*).

**Table 1.** Exploratory factor analysis for cognitive measurements at baseline.
F1 (Cognitive Control) and F2 (Working Memory) refer to the factor loadings of each measure from an exploratory factor analysis with promax rotation ($N$ = 287). Uniqueness represents the variance of each item not accounted for by the two factors.

| Measure | CC | WM | Uniqueness |
|---|---|---|---|
| *Flanker* | 0.89 | | 0.21 |
| *Go/no-go* | 0.71 | | 0.48 |
| *Stroop* | 0.55 | | 0.71 |
| *Backward digit span* | | 0.70 | 0.51 |
| *Backward Corsi blocks* | | 0.27 | 0.91 |
| *Visual 2-back* | | 0.33 | 0.90 |

*Note*: Only factor loadings greater than. 25 are included in the table.

The online version of this article includes the following source data for  Table 1:

Source data 1. Scree plot for the exploratory factor analysis on all cognitive measures.The plot shows the eigenvalues associated with each factor plotted against each factor, and supports the decision to retain two factors.

Source data 2. Path diagram for the exploratory factor analysis on all cognitive measures.F1 (Cognitive Control) and F2 (Working Memory) refer to the factors extracted from an exploratory factor analysis on all six cognitive measures, with promax rotation ($N$ = 287).

## Cognitive improvements

Here, we report cognitive improvements broken down by constructs, defined based on the factors extracted from the exploratory factor analysis.

A Bayesian repeated measures ANOVA on *Cognitive Control* scores, with *Session* (pretest vs. posttest) as a within factor and *Condition* (HIT vs. Control) as a between factor, showed moderate evidence for the interaction model over the main effect model [$BF_M$ = 3.38, $p$(M | Data)=0.46; *Table 2*]. Participants in the HIT group showed larger improvements than controls from pretest to posttest ($M_{gain}$ = 0.25, $SD_{gain}$ = 0.6 and $M_{gain}$ = 0.08, $SD_{gain}$ = 0.47, respectively, Hedges' *g* = 0.31 [0.09, 0.54]; *Figure 2A*, see also Figures 4–6). A Bayesian repeated measures ANOVA on *Working Memory* scores, with *Session* (pretest vs. posttest) as a within factor and *Condition* (HIT vs. Control) as a between factor showed strong evidence for the interaction model over the main effect model [$BF_M$ = 5233.68, $p$(M | Data) ≈1; *Table 3*]. Participants in the HIT group showed larger improvements than controls from pretest to posttest ($M_{gain}$ = 0.48, $SD_{gain}$ = 0.83 and $M_{gain}$ = 0.12, $SD_{gain}$ = 0.44, respectively, Hedges' *g* = 0.54 [0.31, 0.77]; *Figure 2B*).

Because the cognitive improvements we reported are presumably based on physiological changes, we directly tested the relationship between the two types of variables. A Bayesian regression analysis showed that change in resting heart rate was a reliable predictor of cognitive gains in the HIT group, with respect to Cognitive Control ($BF_{10}$ = 6.34, $p$(M | Data)=0.86). This was not the case in the Control group ($BF_{10}$ = 0.20, $p$(M | Data)=0.16). The contrast was weaker when comparing Working Memory gains in the HIT group ($BF_{10}$ = 0.56, $p$(M | Data)=0.36) with those of the Control group ($BF_{10}$ = 0.18, $p$(M | Data)=0.15). Additional Bayesian regression analyses showed that lower resting heart rate at pretest did not predict improvements in either Cognitive Control or Working Memory in the HIT group ($BF_{10}$ = 0.41, $p$(M | Data)=0.29. and $BF_{10}$ = 0.18, $p$(M | Data)=0.15, respectively). This was also the case when the analyses were restricted to the. 75 quantile of individuals with the lowest resting heart rate at baseline ($BF_{10}$ = 0.34, $p$(M | Data)=0.25. and $BF_{10}$ = 0.62, $p$(M | Data)=0.38, respectively). Overall, baseline resting heart rate was a fairly noisy measure ($M$ = 85.2, $SD$ = 14.77, over the entire sample) and this might have contributed to the lack of clear impact of resting heart rate *change* on cognitive function.

## Effect of *BDNF* genotype

A subsample of our data allowed for a better understanding of individual differences in exercise-induced cognitive improvements. Specifically, we looked at the effect of variations in the *BDNF* polymorphism on cognitive gains in the HIT group, via a comparison between met[66] carriers (i.e. met[66]/met[66] or val[66]/ met[66]) and non-carriers (val[66] homozygotes). Separate Bayesian repeated measures ANOVAs on *Cognitive Control* and *Working Memory* scores, with *Session* (pretest vs. posttest) as a within factor and *BDNF polymorphism* (val[66] homozygotes vs. met[66] carriers) as a between factor showed strong evidence for the interaction model in both cases [$BF_M$ = 31.17, $p$(M | Data)=0.89, and $BF_M$ = 675.92, $p$(M | Data)=0.99, for Cognitive Control and Working Memory, respectively, see *Table 4* and *Table 5*]. These findings suggest that met[66] carriers benefited to a greater extent than non-carriers from the exercise intervention (Cognitive Control: $M_{gain}$ = 0.93, $SD_{gain}$ = 1.20 and $M_{gain}$ = 0.05, $SD_{gain}$ = 0.13, Hedges' *g* = 1.36 [0.52, 2.2]; Working Memory, $M_{gain}$ = 0.87,

**Table 2.** Model comparisons for the Cognitive Control construct (CC) with condition as a fixed factor.

The table shows the probability of each model given the data $P$(M | Data), the corresponding Bayes Factor, $BF_{10}$ and the percentage of error. The unconditional probability for each model is 0.2.

| Models | $P$(M | Data) | $BF_M$ | $BF_{10}$ | Error (%) |
|---|---|---|---|---|
| Null | 1.01e −5 | 4.05e −5 | 1 | - |
| Session | 0.43 | 3.06 | 43120.06 | 0.98 |
| Condition | 2.32e −6 | 9.28e −6 | 0.23 | 3.93 |
| Main effects | 0.11 | 0.49 | 11143.87 | 4.52 |
| Interaction | 0.46 | 3.38 | 43792.87 | 5.52 |

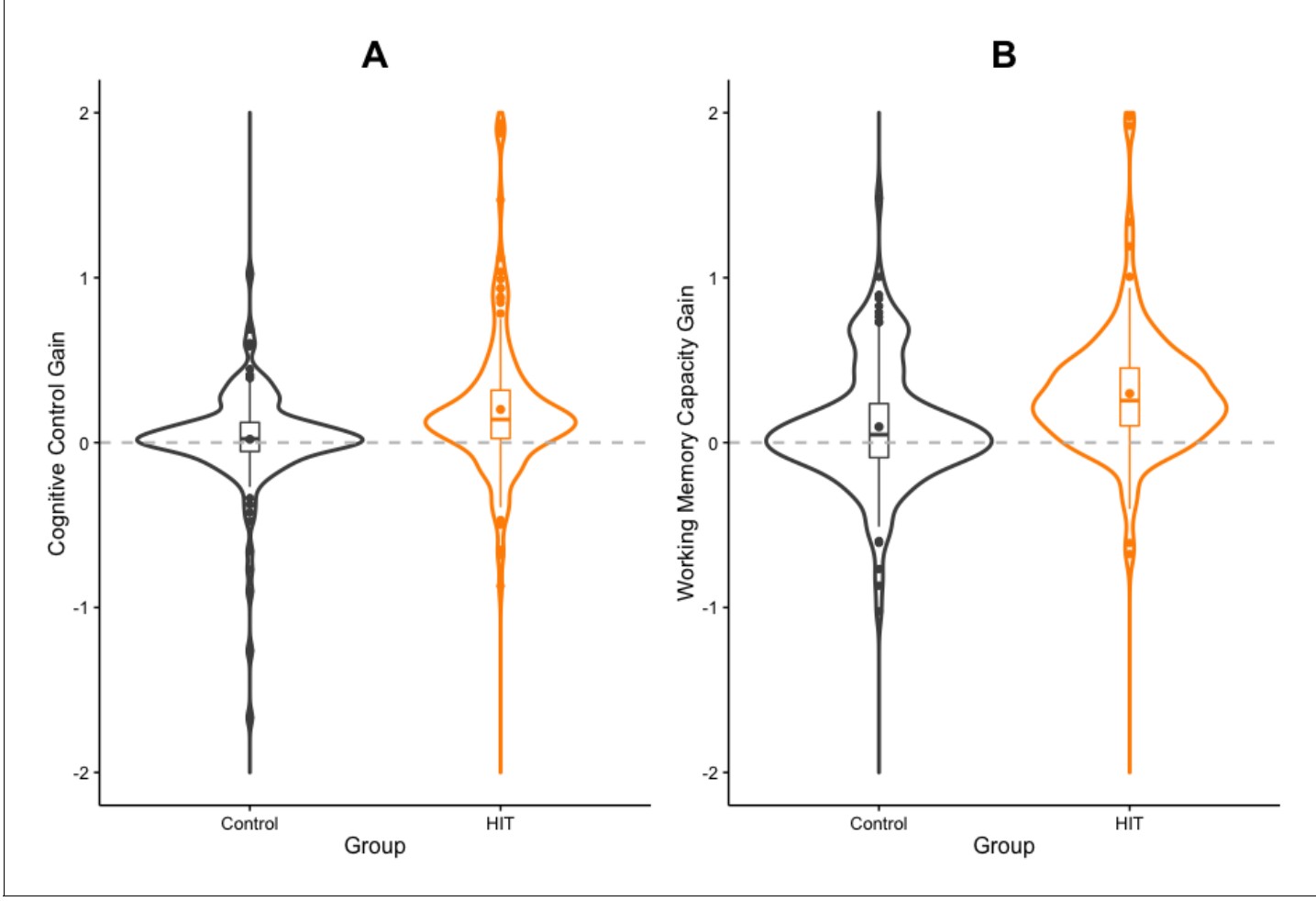

**Figure 2.** Cognitive improvements. Violin and box plots showing gains in Cognitive Control (**A**) and Working Memory (**B**) between pretest and posttest sessions, for HIT and control groups.

$SD_{gain}$ = 0.64 and $M_{gain}$ = 0.14, $SD_{gain}$ = 0.24, Hedges' $g$ = 1.83 [0.94, 2.72]; *Figure 3*). Unequal baseline scores cannot fully account for this effect since evidence for differences in Cognitive Control was limited at pretest ($BF_{10}$ = 4.03, Error (%)=1.55 e −6 from a Bayesian independent samples t-test) and more substantial, but unable to account for the full effect, for Working Memory ($BF_{10}$ = 17.47, Error (%)=1.22 e −6 from a Bayesian independent samples t-test). Together, these

**Table 3.** Model comparisons for the Working Memory construct (WM) with condition as a fixed factor.

The table shows the probability of each model given the data $P$(M | Data), the corresponding Bayes Factor, $BF_{10}$ and the percentage of error. The unconditional probability for each model is 0.2.

| Models | $P$(M | Data) | $BF_M$ | $BF_{10}$ | Error (%) |
|---|---|---|---|---|
| Null | 2.92e −13 | 1.17e −12 | 1 | - |
| Session | 4.49e −4 | 0 | 1.54e + 9 | 1.23 |
| Condition | 1.84e −13 | 17349e −13 | 0.63 | 0.69 |
| Main effects | 3.15e −4 | 0 | 1.08e + 9 | 3.13 |
| Interaction | 1 | 5232.68 | 3.42e + 12 | 2.91 |

**Table 4.** Model comparisons for the Cognitive Control construct (CC) with *BDNF* polymorphism as a fixed factor.
The table shows the probability of each model given the data $P$(M | Data), the corresponding Bayes Factor, $BF_{10}$ and the percentage of error. The unconditional probability for each model is 0.2.

| Models | $P$(M | Data) | $BF_M$ | $BF_{10}$ | Error (%) |
|---|---|---|---|---|
| Null | 0.01 | 0.06 | 1 | - |
| Session | 0.03 | 0.12 | 1.94 | 1.45 |
| Condition | 0.02 | 0.10 | 1.65 | 2.60 |
| Main effects | 0.04 | 0.19 | 3.05 | 1.71 |
| Interaction | 0.89 | 31.17 | 59.49 | 24.61 |

findings indicate that although genetic variations in the *BDNF* polymorphism are associated with cognitive differences, the latter are malleable and can be reduced with physical exercise.

## Priors and robustness

All priors used in the reported analyses are default prior scales (*Morey and Rouder, 2015*). For Bayesian repeated measures ANOVA and ANCOVA, the prior scale on fixed effects is set to 0.5, the prior scale on random effects to 1, and the prior scale on the covariate to 0.354. The latter is also used in Bayesian Linear Regression. The Bayesian t-test uses a Cauchy prior with a width of $\sqrt{2}/2$ (~0.707), that is half of parameter values lies within the interquartile range $[-0.707; 0.707]$.

It is worth pointing out that the Bayesian repeated measures ANOVA that showed only moderate evidence for the effect of our HIT intervention on Cognitive Control shows stronger evidence with a slight variation on the prior scale. Although this variation in priors is consistent with our data and provides stronger evidence for our claim, we chose to report analyses with default prior scales, as these were the intended parameters *a priori.* For transparency, we plotted below the prior and posterior distribution for the comparison between *Conditions* (HIT vs. Control) for Cognitive Control (*Figure 4*), as well as the Bayes Factor robustness check (*Figure 5*). Both indicate that our findings are robust and supported by a wide range of priors, as corroborated by a sequential analysis (*Figure 6*).

## Markov chain Monte Carlo (MCMC) Parameters

Broadly speaking, MCMC methods approximate the true posterior density $p(\theta \mid y)$ by constructing a Markov chain on the state space $\theta \in \Theta$. The probability of the subsequent state in a given chain can be defined as:

$$P(X_{n+1} = i_{n+1} | X_n = I_n), \ \ I \ \in \ \theta \tag{2}$$

where $\{X_0, X_1,..\}$ is a sequence of random variables and $\theta$ is the state space. Accordingly, the state

**Table 5.** Model comparisons for the Working Memory construct (WM) with *BDNF* polymorphism as a fixed factor.
The table shows the probability of each model given the data $P$(M | Data), the corresponding Bayes Factor, $BF_{10}$ and the percentage of error. The unconditional probability for each model is 0.2.

| Models | $P$(M | Data) | $BF_M$ | $BF_{10}$ | Error (%) |
|---|---|---|---|---|
| Null | 2.47e −5 | 9.90e −5 | 1 | - |
| Session | 0 | 0.01 | 79.86 | 0.77 |
| Condition | 3.52e −5 | 1.41e −4 | 1.42 | 0.69 |
| Main effects | 0 | 0.01 | 155.37 | 5.84 |
| Interaction | 0.99 | 675.92 | 40159.39 | 3.51 |

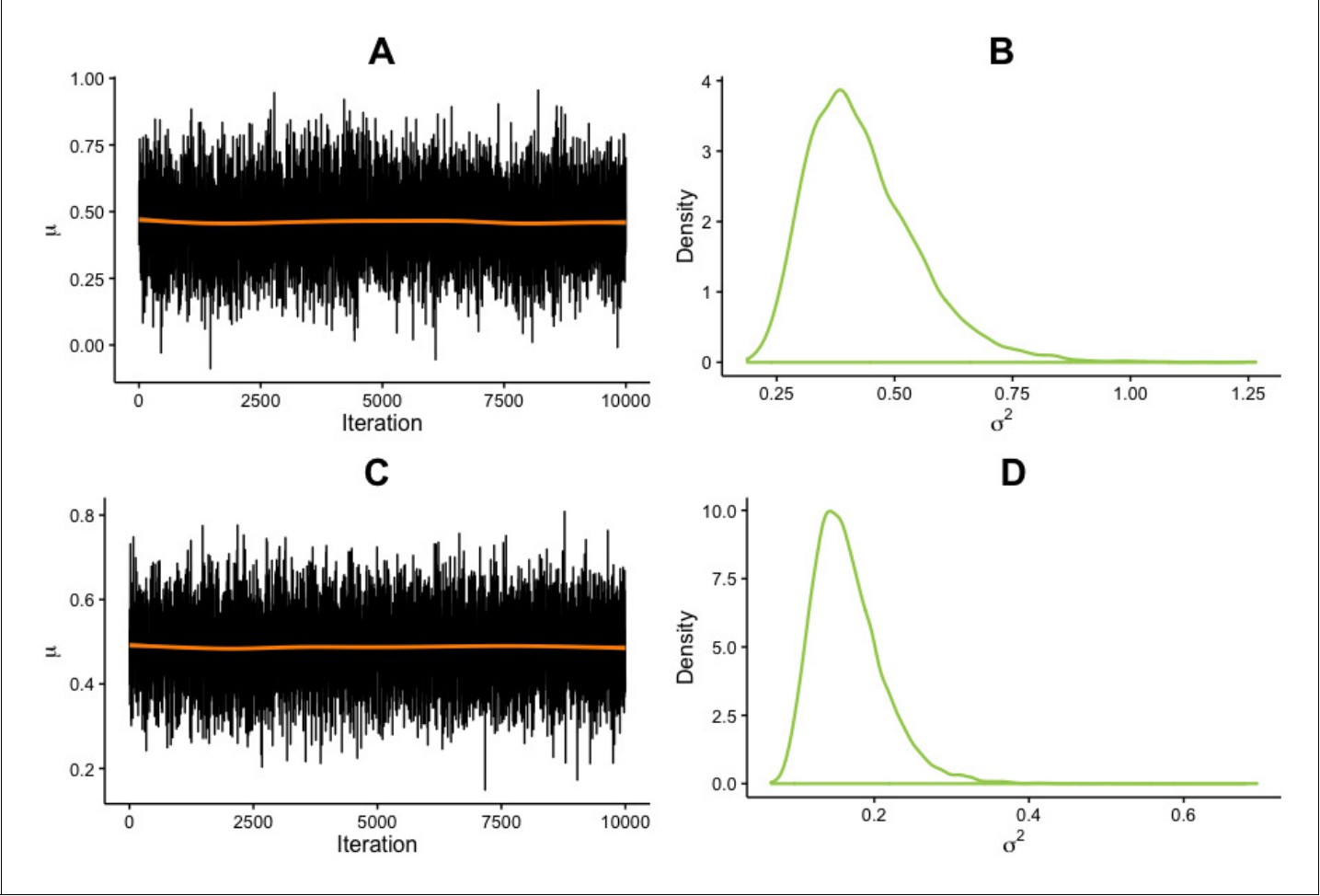

**Figure 3.** Effect of *BDNF* allele on cognitive improvements. $\mu$ and $\sigma^2$ parameter estimates from the posterior distribution for the difference between *BDNF* met carriers and non-carriers (met[66] – val[66] homozygotes) in cognitive gains. Estimates were generated from 10,000 iterations, in one chain, with thinning interval of one (no data point discarded). (**A**) Trace of $\mu$ for Cognitive Control. (**B**) $\sigma^2$ estimate for Cognitive Control. (**C**) Trace of $\mu$ for Working Memory. (**D**) $\sigma^2$ estimate for Working Memory.

at time step *n + 1* is dependent only on the state at time *n*. This process is best represented with a random walk where each vertex is defined by $\theta$, and weighted by the transition probabilities:

$$p_{ij} = P(X_{n+1} = j | X_n = i), \; i,j \in \theta \tag{3}$$

In the analyses reported in the paper, MCMC was used to generate posterior samples via the Metropolis-Hastings algorithm (see for details *Rubinstein and Kroese, 2011*). All analyses were set at 10,000 iterations, with diagnostic checks for convergence. One chain per analysis was used for all analyses reported in the paper, with a thinning interval of 1 (i.e., no iteration was discarded).

## Frequentist analyses

We reported Bayesian analyses throughout the paper. Because we understand that some readers may wish to compare these results with the equivalent frequentist analyses, we are providing all of these herein, in the order of presentation in the paper. Note that an *a priori* power analysis based on previous studies (*Erickson et al., 2013*; *Moreau et al., 2015*) indicated the need for a minimum *N* of 129 participants per group to detect an effect of *d* = 0.35 on the primary outcome measures, with 1 − $\beta$ = 0.80 and $\alpha$ = 0.05. The actual sample size of the present study (*N* = 152 and *N* = 153, for HIT and control groups, respectively) allowed an *a priori* power of. 86, given *d* and $\alpha$ constant.

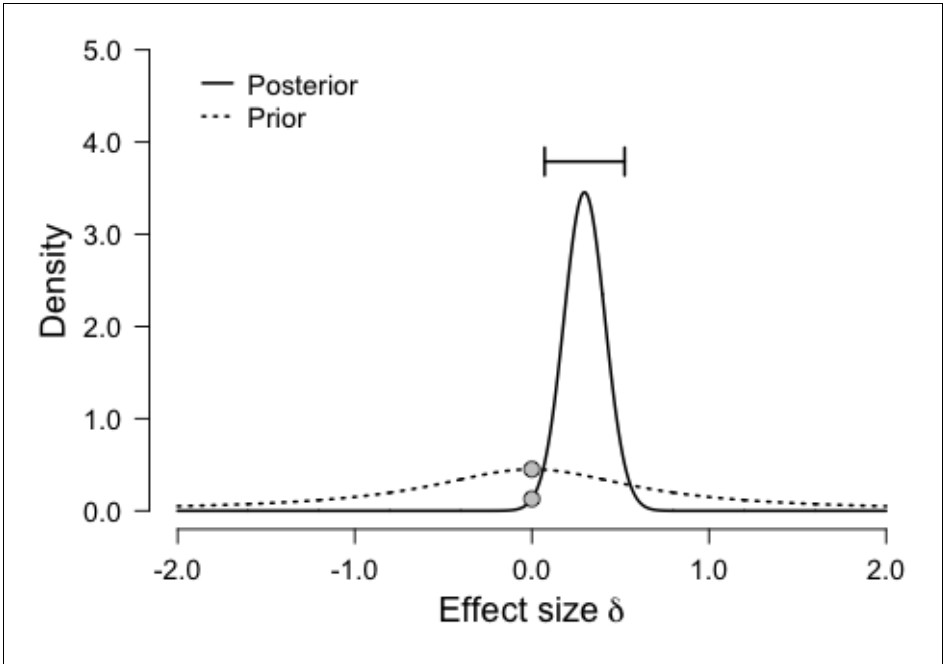

**Figure 4.** Prior and posterior distributions for the comparison between *Conditions* (HIT vs. Control) for Cognitive Control. The graph shows the density of each distribution as a function of effect size, with the prior centered on the null effect.

- ANCOVA on change in resting heart rate, with *Condition* (HIT vs. Control) as a fixed factor and baseline heart rate as a covariate. Main group effect: $F(1, 301)=9.84$, p=0.002, $\eta^2 = 0.02$ (.00,. 08), *Figure 1A*. Levene's test for homogeneity of variance was not statistically significant: $F(1, 302)=0.29$, p=0.59.
- Welch two-sample t-test on resting heart rate change, by *Condition* (HIT vs. Control), after split on resting heart rate at pretest: $t(1, 146.4)=2.57$, p=0.01. $M_{gain} = 6.11$, $SD_{gain} = 11.64$ and $M_{gain} = 1.89$, $SD_{gain} = 8.63$, for HIT and Control, respectively; Hedges' $g = 0.41$ (0.09, 0.73).
- Linear regressions showed sustained effort across training sessions ($\beta_1 10.04$, p>0.05; *Figure 1C*), but incremental workout load ($\beta_1 18.81$, p<0.01; *Figure 1D*).
- Repeated measures ANOVA on *Cognitive Control* scores, with *Session* (pretest vs. posttest) as a within factor and *Condition* (HIT vs. Control) as a between factor. Interaction effect: $F(1, 295)=7.17$, p=0.008, $\eta^2 = 0.02$ (.00,. 07). Participants in the HIT group showed larger improvements than controls from pretest to posttest ($M_{gain} = 0.25$, $SD_{gain} = 0.6$ and $M_{gain} = 0.08$, $SD_{gain} = 0.47$, respectively, Hedges' $g = 0.31$ [0.09, 0.54]). Levene's test for homogeneity of variance was not statistically significant: $F(1, 295)=0.35$, p=0.56 and $F(1, 295)=0.82$, p=0.37, at pretest and posttest respectively.
- Repeated measures ANOVA on *Working Memory* scores, with *Session* (pretest vs. posttest) as a within factor and *Condition* (HIT vs. Control) as a between factor. Interaction effect: $F(1, 287)=21.89$, p<0.001, $\eta^2 = 0.06$ (.00,. 11). Participants in the HIT group showed larger improvements than controls from pretest to posttest ($M_{gain} = 0.48$, $SD_{gain} = 0.83$ and $M_{gain} = 0.12$, $SD_{gain} = 0.44$, respectively, Hedges' $g = 0.54$ [0.31, 0.77]). Levene's test for homogeneity of variance was not significant at pretest: $F(1, 287)=0.16$, p=0.69, but was statistically significant at posttest: $F(1, 287)=19.75$, p<0.001.
- Regression analysis on *Cognitive Control* gains with change in resting heart rate as a predictor: HIT, $F(1, 149)=7.93$, p<0.01, $R^2 = 0.05$, RMSE = 0.58. Control group, $F(1, 144)=0.24$, p=0.63, $R^2 = 0.002$, RMSE = 0.48.
- Regression analysis on *Working Memory* gains with change in resting heart rate as a predictor: HIT, $F(1, 137)=2.47$, p=0.12, $R^2 = 0.02$, RMSE = 0.82. Control group, $F(1, 147)=0.08$, p=0.77, $R^2 = 0.001$ RMSE=0.44.

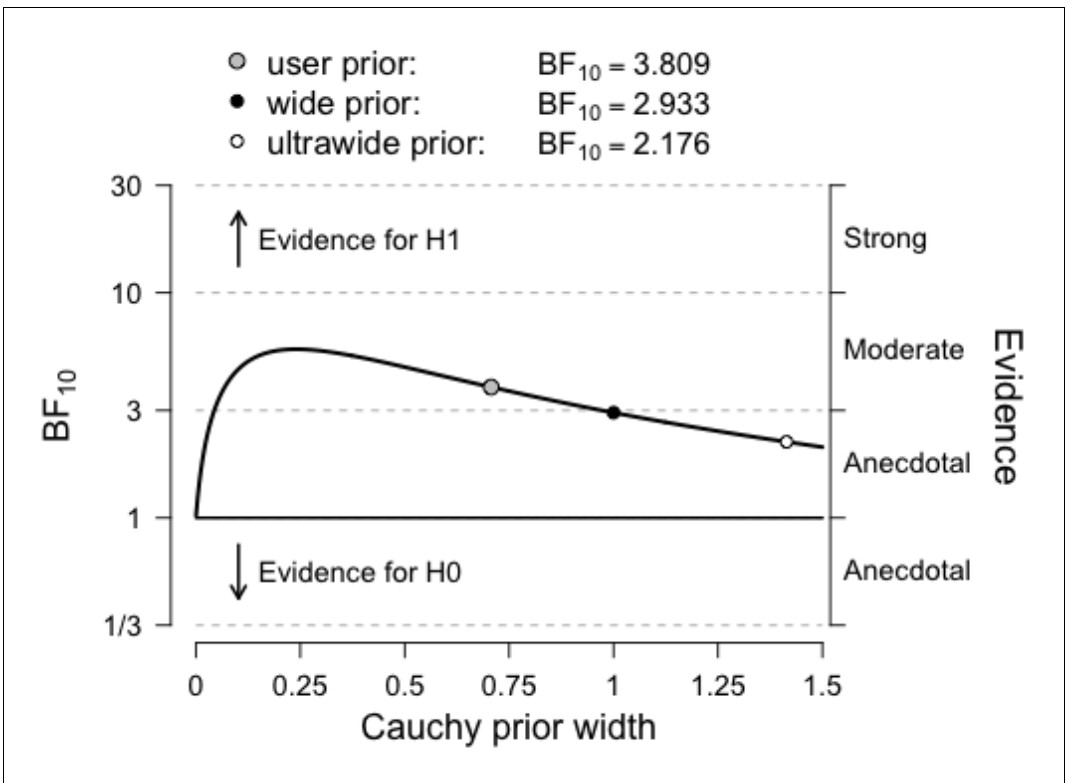

**Figure 5.** Bayes factor robustness check for the comparison between *Conditions* (HIT vs. Control) for Cognitive Control. The figure shows our default prior, as well as wide and ultrawide priors. Importantly, the curve shows stronger evidence for our hypothesis with narrower priors, indicating that our conclusions are not based on a restricted range of priors.

- Regression analysis on *Cognitive Control* gains with baseline resting heart rate as a predictor (HIT group only): $F(1, 149)=1.83$, p=0.18, $R^2 = 0.01$, RMSE = 0.59. When restricted to the. 75 quantile: $F(1, 44)=0.37$, p=0.54, $R^2 = 0.01$, RMSE = 0.73.
- Regression analysis on *Working Memory* gains with baseline resting heart rate as a predictor (HIT group only): $F(1, 137)=0.02$, p=0.90, $R^2 = 0$, RMSE = 0.83. When restricted to the. 75 quantile: $F(1, 40)=1.79$, p=0.19, $R^2 = 0.04$, RMSE = 0.40.
- Repeated measures ANOVA on *Cognitive Control* scores, with *Session* (pretest vs. posttest) as a within factor and *BDNF polymorphism* (val vs. met) as a between factor. Interaction effect: $F(1, 30)=12.73$, p=0.001, $\eta^2 = 0.22$ (.00,. 50). Met[66] carriers benefited to a greater extent than non-carriers from the exercise intervention (respectively $M_{gain} = 0.93$, $SD_{gain} = 1.20$ and $M_{gain} = 0.05$, $SD_{gain} = 0.13$, Hedges' $g = 1.36$ [0.52, 2.2]). Levene's test for homogeneity of variance was statistically significant a pretest: $F(1, 30)=25.20$, p<0.001, but not at posttest $F(1, 30)=2.16$, p=0.15.
- Repeated measures ANOVA on *Working Memory* scores, with *Session* (pretest vs. posttest) as a within factor and *BDNF polymorphism* (val vs. met) as a between factor. Interaction effect: $F(1, 30)=23.01$, p<0.001, $\eta^2 = 0.24$ (.01,. 47). Met[66] carriers benefited to a greater extent than non-carriers from the exercise intervention (respectively $M_{gain} = 0.87$, $SD_{gain} = 0.64$ and $M_{gain} = 0.14$, $SD_{gain} = 0.24$, Hedges' $g = 1.83$ [0.94, 2.72]). Levene's test for homogeneity of variance was statistically significant a pretest: $F(1, 30)=14.48$, p<0.001, but not at posttest $F(1, 30)=0.33$, p=0.57.
- Welch two-sample t-test on *Cognitive Control* score at pretest, by *BDNF* polymorphism (val[66] vs. met[66]): $t(1, 8.3)=1.68$, p=0.13, Hedges' $g = 1.01$ (0.20, 1.81).
- Welch two-sample t-test on Working Memory score at pretest, by *BDNF polymorphism* (val[66] vs. met[66]): $t(1, 8.7)=2.30$, p<0.05, Hedges' $g = 1.31$ (0.47, 2.14).

We present below analyses for each cognitive tasks included in this study. Descriptive statistics are reported in *Table 6*.

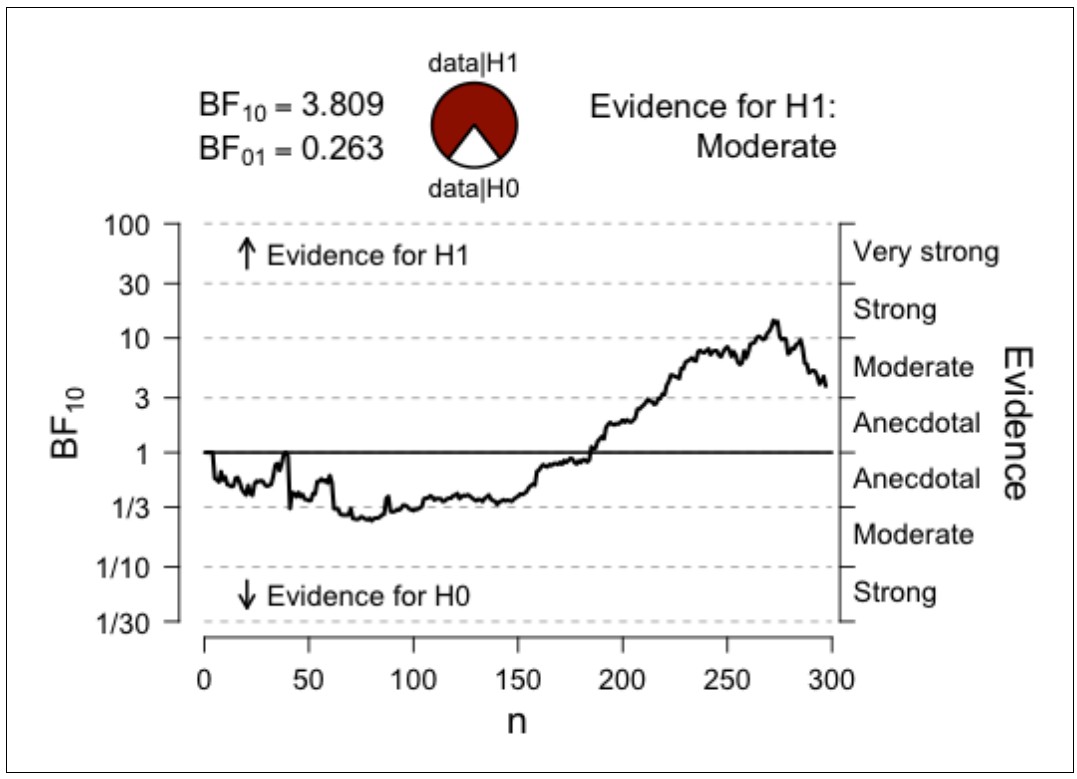

**Figure 6.** Sequential analysis. The graph shows the strength of evidence (as expressed by $BF_{10}$) as $N$ increases.

- Repeated measures ANOVA on *Flanker* scores, with *Session* (pretest vs. posttest) as a within factor and *Condition* (HIT vs. Control) as a between factor. Interaction effect: $F(1, 301)=4.00$, $p<.05$, $\eta^2 = 0.01$ (.00,. 05). Levene's test for homogeneity of variance was not statistically significant: $F(1, 301)=0.38$, $p=0.56$ and $F(1, 301)=1.14$, $p=0.29$, at pretest and posttest respectively.
- Repeated measures ANOVA on *Go/no-go* scores, with *Session* (pretest vs. posttest) as a within factor and *Condition* (HIT vs. Control) as a between factor. Interaction effect: $F(1, 301)=4.52$, $p=0.03$, $\eta^2 = 0.01$ (.00,. 05). Levene's test for homogeneity of variance was not statistically significant: $F(1, 297)=1.24$, $p=0.26$ and $F(1, 297)=0.02$, $p=0.88$, at pretest and posttest respectively.
- Repeated measures ANOVA on *Stroop* scores, with *Session* (pretest vs. posttest) as a within factor and *Condition* (HIT vs. Control) as a between factor. Interaction effect: $F(1, 301)=2.63$, $p=0.11$, $\eta^2 <0.01$ (.00,. 04). Levene's test for homogeneity of variance was not statistically

**Table 6.** Mean cognitive scores (*SDs*) for the two conditions at pretest and posttest. Scores are scaled and centered for each task (z-transformed by row).

|  | HIT | | Control | |
| --- | --- | --- | --- | --- |
|  | **Pretest** | **Posttest** | **Pretest** | **Posttest** |
| *Flanker* | −0.14 (*1.20*) | 0.16 (*0.66*) | −0.06 (*1.18*) | 0.04 (*0.85*) |
| *Go/no-go* | −0.09 (*1.11*) | 0.08 (*0.96*) | 0.01 (*1.04*) | 0.01 (*0.88*) |
| *Stroop* | −0.11 (*1.19*) | 0.16 (*0.38*) | −0.09 (*1.31*) | 0.04 (*0.83*) |
| *Backward digit span* | −0.14 (*1.07*) | 0.25 (*0.55*) | −0.13 (*1.34*) | 0.02 (*0.82*) |
| *Backward Corsi blocks* | −0.13 (*1.55*) | 0.31 (*0.35*) | −0.17 (*0.92*) | 0.00 (*0.73*) |
| *Visual 2-back* | −0.24 (*1.62*) | 0.33 (*0.51*) | −0.06 (*0.74*) | −0.03 (*0.69*) |

significant: $F(1, 301)=0.05$, p=0.83 and $F(1, 301)=2.53$, p=0.11, at pretest and posttest respectively.

- Repeated measures ANOVA on *Backward digit span* scores, with *Session* (pretest vs. posttest) as a within factor and *Condition* (HIT vs. Control) as a between factor. Interaction effect: $F(1, 301)=5.66$, p=0.02, $\eta^2 = 0.02$ (.00,. 06). Levene's test for homogeneity of variance was not statistically significant at pretest but was statistically significant at posttest: $F(1, 301)=0.63$, p=0.43 and $F(1, 301)=6.41$, p=0.01, respectively.
- Repeated measures ANOVA on *Backward Corsi blocks* scores, with *Session* (pretest vs. posttest) as a within factor and *Condition* (HIT vs. Control) as a between factor. Interaction effect: $F(1, 297)=4.49$, p=0.03, $\eta^2 = 0.01$ (.00,. 05). Levene's test for homogeneity of variance was not statistically significant at pretest but was statistically significant at posttest: $F(1, 297)=0.07$, p=0.79 and $F(1, 297)=34.69$, p<0.001, respectively.
- Repeated measures ANOVA on *Visual 2-back* scores, with *Session* (pretest vs. posttest) as a within factor and *Condition* (HIT vs. Control) as a between factor. Interaction effect: $F(1, 292)=16.22$, p<0.001, $\eta^2 = 0.05$ (.02,. 11). Levene's test for homogeneity of variance was not statistically significant at pretest but was statistically significant at posttest: $F(1, 292)=2.27$, p=0.13 and $F(1, 292)=16.41$, p<0.001, respectively.

## Additional analyses

Here, we report analyses for which our *a priori* hypotheses were null effects. These variables were collected either to control for potential confounds, or for exploratory purposes.

There was no difference between groups regarding self-reported enjoyment or motivation ($W$ = 12058, p=0.54 and $W$ = 11497, p=0.86, respectively). This finding allows controlling for expectation effects, and thus stronger causal claims (*Boot et al., 2013*; *Stothart et al., 2014*). In addition, participants' self-reported belief about cognitive malleability (i.e., mindset) indicated a statistically significant difference (p<0.03) in favor of the control group ($M$ = 7.11, SD = 2.65 and $M$ = 6.42, SD = 2.74, respectively, Hedges' $g$ = 0.26 [0.03, 0.48]). There was no statistically significant difference between groups at either time points (pretest or posttest) in terms of ethnic background, age, gender, handedness, height, weight, diagnosis of learning disorder, brain trauma or epileptic seizures, current or past enrolment in a remediation or a cognitive training program, English as first language, videogaming experience, physical exercise, self-reported happiness, sleep quality, or general health.

## Discussion

The present study reported the first experimental evidence that HIT can elicit robust cognitive improvements in children. We confirmed the main hypothesis that exercise could induce gains in both cognitive control and working memory, as assessed from multiple measures. This finding is particularly promising given that the two constructs are reliable predictors of success in many domains, including professional and academic (*Deary et al., 2007*); in the classroom, cognitive control and working memory have been associated with effective learning and overall achievement (*Rohde and Thompson, 2007*). Importantly, these effects are also meaningful at the level of single tasks. Our main findings thus emphasize the potency of short but intense exercise interventions to enhance cognition, and suggests that aerobic exercise is not the sole means to elicit cognitive gains, in line with a growing body of research (*Liu-Ambrose et al., 2012*; *Moreau et al., 2015*; *Pesce et al., 2016*; *Tomporowski et al., 2015b*). HIT appears to be a viable and promising alternative to longer workouts to enhance cognition.

In addition to the main effect of training, we also postulated that specific genetic profiles would be correlated with different responses to training, with *BDNF* met[66] carriers (i.e. met/met or val/met) benefiting from exercise to a greater extent. This hypothesis, confirmed by the present findings, was based on previous literature showing a relationship between *BDNF* polymorphism and serum *BDNF* (*Lang et al., 2009*), and the influence of exercise interventions on the latter (*Leckie et al., 2014*). *BDNF* met[66] carriers showed greater gains from pretest to posttest on both cognitive constructs. As the substitution from valine to methionine at codon 66 typically results in decreases of activity-dependent secretion of *BDNF* at the synapse (*Egan et al., 2003*), *BDNF* met[66] carriers are thought to be particularly impacted by post-exercise *BDNF* increases (*Nascimento et al., 2015*). Conversely, val[66] homozygotes might benefit less from *BDNF* increases given above-average baseline levels.

This finding is interesting because of its predictive power—controlling for *BDNF* polymorphism can allow more accurate forecasting of individual training responses, and better estimates of effect size. Given the current trend toward more personalized interventions (*Medalia and Richardson, 2005*; *Moreau and Waldie, 2015*), factoring in genetic information has the potential to refine and improve regimens, for each individual.

Importantly, the main effect of exercise on cognitive function is unlikely to be explained by the placebo effect. Self-reported measures of enjoyment and motivation did not differ between groups, supporting the notion that both types of intervention were equally appealing to children, or at least not fundamentally different with respect to intrinsic motivating factors. In addition, we also controlled for mindset—the degree to which individuals believe their cognitive abilities can change over time (*Dweck et al., 1995*). Previous research suggests that individual mindsets may be of influence in cognitive growths: individuals with a more 'malleable' mindset are thought to be more likely to improve in cognitive and academic domains than individuals with a 'fixed' mindset (*Paunesku et al., 2015*). Although we did find a difference between conditions, it was to the advantage of children in the control group, who held a more malleable view about the dynamic properties of cognitive function than did the HIT group. This finding was reported as an additional analysis in the Results section, rather than in the main section, because the effect was not hypothesized *a priori*. In any case, and with warranted caution regarding *post hoc* interpretation, this effect would suggest that controls were *more* likely to improve over time, an assumption that was not corroborated by our main finding. If anything, this strengthens our main claim—greater improvements in the HIT condition are inconsistent with a differential placebo effect, a critical point in light of recent findings in the field of cognitive training (*Foroughi et al., 2016*).

Effort-dependent variables provided additional insight about the mechanism underlying improvements. Consistent with previous interventional studies that have investigated the influence of exercise on cognition, we actively monitored several variables throughout the intervention. We could thus ensure that training was adequate, performed at a suitable intensity, and could test directly the dynamic coupling of these variables and their effects on cognitive outcomes. Indeed, we found that almost all participants stayed within the overall targeted range of effort required by the design of the workout and by initial individual measurements. This indicates a high degree of agreement, or fidelity, to the intended protocol—an essential component of the intervention given the underlying assumption that participants would exercise at a high intensity. Performed at a more moderate intensity, the same regimen becomes an aerobic training program, for which substantially longer time commitment might be required to elicit improvements. In addition, it is important to note that participants incrementally increased workout load, thus maintaining appropriate intensity throughout the intervention. Arguably, this adaptive property emerged from the design of the intervention, whereby participants were encouraged to exercise at maximum intensity at the time of the workout—an intrinsically individual and dynamic variable by definition.

In the present study, the effect of exercise could also be appreciated on changes in resting heart rate from pretest to posttest, a finding that confirms previous research in the field (e.g., *Moreau et al., 2015*). More specifically, exercise appears to be a valuable regulating mechanism, with elevated resting heart rate values likely to normalize as a result of HIT. This effect could not be attributed to regression toward the mean, as it was not found in controls, thus suggesting that the intervention is especially beneficial to individuals who need it most. Together with the finding that HIT benefited more individuals with a genetic polymorphism (*BDNF* met[66] carriers) that was associated with lower cognitive performance, this idea emphasizes the relevance of exercise interventions to individuals with specific genetic or physiological attributes to reduce interindividual differences (*Gómez-Pinilla et al., 2001*; *Leckie et al., 2014*). Disparities are genuine, yet targeted interventions allow low-performing individuals to improve dramatically.

Our design allowed delving deeper into the relationship between exercise and cognitive improvements. In particular, we found that change in resting heart rate was predictive of cognitive control, but not working memory, gains. This finding is of interest because we postulated that the mechanisms of improvement were physiological, given that HIT has been shown to elicit neurophysiological changes similar to those following aerobic exercise regimens (*Ferris et al., 2007*). However, we should point out that baseline resting heart rate did not predict gains in cognitive control or working memory in the HIT group, emphasizing the inherent noise associated with the relationship between physiological and cognitive measures. This lack of clear association between both types of variables

is fairly common in the literature, suggesting that the hypothesized mechanisms of improvements are difficult to elucidate (*Moreau et al., 2015*; *Tsai et al., 2014*). Potentially, this also suggests that other variables may be of importance, a question we attempted to address with additional measurements, in an exploratory manner (see Additional Analyses in the Results section).

Regardless of the strength of the evidence reported in the present study, one might question the genuine impact of such a short training regimen. Although perhaps counterintuitive, the extreme potency of short, intense bursts of exercise has come to light in recent years (*Lucas et al., 2015*; *Rognmo et al., 2004*). In a review of the impact of HIT on public health, *Biddle and Batterham, 2015* go as far as to premise their entire argument on the idea that the cardio-metabolic health outcomes are not to be questioned—rather, their only concern was about whether or not such regimens can be widely implemented and sustained over time (*Biddle and Batterham, 2015*). In a similar vein, Costigan and colleagues concluded that HIT is a time-efficient approach for improving cardio-respiratory fitness in adolescent populations (*Costigan et al., 2015*). These are strong, unequivocal statements, which reflect current views in exercise physiology—HIT has tremendous health benefits, with little, if any, disadvantages.

More direct evidence for the impact of HIT on brain function comes from neurophysiological studies. In an experiment that directly assessed the impact of short bursts of exercise on BDNF levels, Ferris et al. found that exercise leads to BDNF increases, and that the magnitude of the increase is intensity-dependent (*Ferris et al., 2007*). This result emphasizes the importance of controlling exercise intensity in HIT studies, given that the main determinant of improvement appears to be the intensity of the workout. Indeed, previous studies have looked at the effect of short, but not intense, bursts of exercise on cognition, and found no clear evidence of improvement (*Craft, 1983*). The high-intensity component of this type of exercise regimen is intended to allow for higher workout intensity than traditional workouts, despite shorter overall volume. The brevity of exercise, on the other hand, is simply a byproduct of intensity—one cannot maintain a near-maximal exercise intensity for long periods of time, given that this type of regimen depletes energetic resources rapidly (*Parolin et al., 1999*). In terms of practical implications, this aspect is critical, as it allows designing shorter, more potent workouts.

Despite our findings being in line with previous literature showing that short bursts of exercise can elicit potent cognitive improvements in children (*Piepmeier et al., 2015*; *Pontifex et al., 2013*), and, more generally, with a wider, more general literature linking physical exercise and cognition in children (*Jackson et al., 2016*; *Sibley and Etnier, 2003*), we should also point out a few limitations of the present study. These open up interesting avenues and directions for future research. First, the duration of training was not experimentally manipulated, and therefore the specific question of dose-dependence was not directly assessed. Therefore, we cannot claim that a 6 week regimen is optimal—larger cognitive improvements could possibly be elicited with longer durations, or, alternatively, similar improvements could be induced with a shorter intervention. Similarly, no follow-up tests were performed to assess durability of improvements post-training; although it should be noted that maintaining benefits is arguably less important with short, potent interventions such as the HIT regimen we proposed. In addition, both the duration of the intervention and the specific experimental protocol were constrained by external factors (e.g. feasibility, academic schedule). With respect to the potency of the intervention, however, this is also promising—as little as six weeks of training can induce noticeable improvements, with possible larger effects if physical exercise is sustained.

Related to this idea, we had to work around constraints typically imposed by interventional studies; namely, the necessity to keep testing sessions time-efficient. Beyond logistic considerations, this was also intended to minimize the influence of cognitive fatigue on our results. Despite time constraints, we aimed to preserve testing diversity (i.e., number of tasks per construct) for a few reasons. First, we strived to provide estimates of constructs that minimize task-specific components and extract meaningful domain-general scores. In psychometric testing of working memory capacity, Foster and colleagues demonstrated that the majority of the variance explained by a single WM task is accounted for in the first few blocks, and that the predictive nature of the task remains largely unchanged for practical purposes when tasks are shortened (*Foster et al., 2015*). Second, simulation studies have shown that incorporating more tasks within constructs leads to a better signal-to-noise ratio, resulting in more meaningful measures of an underlying ability (*Moreau et al., 2016*). To further validate this approach, we piloted different versions of our testing tasks and determined that

the validity of the versions we retained for the present study was acceptable, given appropriate reliability across task lengths.

Second, because training occurred in schools, the environment was possibly less standardized and controlled than laboratory settings, highlighting typical tradeoffs between impoverished but highly reliable environments and ecological but less controlled settings. Our view remains that ecological validity is of primary importance when training cognitive abilities, because of the wide range of applications stemming from this line of research (*Moreau and Conway, 2014*). Accordingly, fixed, predictable training regimens are unlikely to favor durable improvements of cognitive function (*Moreau and Conway, 2014*; *Moreau et al., 2015*; *Posner et al., 2015*). We have previously stressed the importance of novelty and variety in cognitive training interventions (*Moreau and Conway, 2014*), and this limitation applies to exercise regimens as well. Importantly, this echoes similar views across research groups worldwide, which suggest that training-induced cognitive improvements are often restricted to specific activities (*Harrison et al., 2013*; *Tomporowski et al., 2012*), and are best nurtured within complex, dynamic environments (*Diamond and Lee, 2011*). Therefore, the regimen we have presented in this paper constitutes a potent short-term intervention, but more variety might be required to elicit long-lasting improvements. In a field that has suffered from setbacks such as lack of replication (e.g. *Redick et al., 2013*; *Thompson et al., 2013*) or common methodological flaws (*Moreau et al., 2016*), it is wise to remain cautious about preliminary studies and emphasize the need for replication.

Despite these limitations, the present findings represent a promising first step toward reliable and affordable exercise-based cognitive interventions, highlighting effective alternatives to aerobic exercise. Together with complementary findings (e.g. *Moreau et al., 2015*), the type of physical exercise regimen we described in this paper could pave the way for novel exercise interventions particularly suited to school environments, which are often constrained by time and equipment. The high-intensity workout we designed did not require any special equipment or any instructors training, and each 10 min session was all-inclusive with warm-up and stretching. Therefore, this type of regimen could also be generalized to other populations; for instance, individuals whose schedule allows little time for exercise, or those who do not intrinsically enjoy exercising, could appreciate opportunities to shorten workouts while preserving the typical benefits of exercise. Adaptations to older populations also represent interesting opportunities considering the benefits typically associated with exercise regimens in these communities (*Colcombe and Kramer, 2003*). The present regimen might require adjustments, given that practicality in a specific context (e.g. managing time constraints in school or professional schedules) potentially differs from practical considerations in another (e.g. mitigating risks of injury in older populations). In any case, generalization is of the rationale, not necessarily of the specific workout that was designed for this intervention.

Finally, it is important to acknowledge that physical exercise, regardless of the specific training regimens considered, is not a panacea when it comes to addressing cognitive deficits—in some cases, especially in the presence of specific conditions or disorders, more targeted or individualized interventions might be required (e.g., *Moreau and Waldie, 2015*), and the ability for exercise regimens to remediate core cognitive deficits might appear inherently limited. However, it remains that physical exercise is one of the most potent and wide-ranging means currently available to enhance cognition non-invasively, with a myriad of positive side effects.

## Materials and methods

We report here a multicenter, randomized (1:1 allocation), placebo-controlled trial. Design and reporting are consistent with CONSORT guidelines (http://www.consort-statement.org/). Participants, parents and school principals gave their informed consent for inclusion in this study, and the Ethics Committee at the University of Auckland approved all procedures. The full protocol and statistical analysis plan are available online at https://github.com/davidmoreau/2017_eLife.

### Participants

A total of 318 children participated in this study. Thirteen participants were not included in the analyses because of dropouts ($N = 7$), extensive missing data ($N = 4$) or problems in data collection ($N = 2$, see CONSORT flow diagram for details). Our final sample consisted of 305 children ($M_{age} = 9.9$ (7–13), $SD_{age} = 1.74$, 187 female, $M_{BMI} = 18.3$, $SD_{BMI} = 6.26$). They were recruited from

six schools across New Zealand, providing a sample of various socioeconomic deciles (three public institutions, three private), locations (three urban institutions, three rural), and ethnic backgrounds representative for the country (70% New Zealand European, 20% Pacific, 7% Asian, 3% Other). The number of students involved per school ranged from 5 to 83 ($M$ = 50.8, $SD$ = 31). All participants reported no history of brain trauma or epilepsy, and all had self-reported normal or corrected-to-normal vision. A subset of 22 children reported a learning disability diagnosis (dyslexia: 14, ADHD: 3, Autism spectrum disorder: 3, mild developmental delay: 2, Irlen syndrome: 2, dyscalculia: 1, dyspraxia: 1). Respective subsets of 284, 99 and 32 participants underwent all assessments, measurements and genotyping described below. All the variables measured in the experiment are reported hereafter.

## Cognitive assessments

Testing was conducted on school premises. All cognitive assessments were computer-based, administered in groups of a maximum of 15 students. This limit on the number of participants tested at a given time was implemented to minimize potential averse effects of group testing. These assessments have shown to be adequate measures of both cognitive control and working memory (*Anderson-Hanley et al., 2012*; *Aron and Poldrack, 2005*; *Kane et al., 2004*; *Nee et al., 2007*; *Pajonk et al., 2010*; *Rudebeck et al., 2012*; *Unsworth and Engle, 2007*). For each task, we measured accuracy and response time. Different stochastic variations of all tasks were used at pretest and posttest. Unless specified otherwise, the number of trials varied based on individual performance to allow reaching asymptotes, with a minimum and a maximum specified for each task. The reliability of this method for each task was assessed from a separate sample ($N$ = 34, $M_{age}$ = 10.3 (8–12), 15 females), and deemed acceptable (all $\rho s$ > 0.65) based on Spearman-Brown prophecy formula (*Brown, 1910*; *Spearman, 1910*). Specifically, reliability was calculated by comparing test scores on the asymptotic version vs, the maximal-length version, for each task (see trial length details below and online repository for source code data). The order below was the order of presentation for every participant at both pretest and posttest (i.e., Flanker – Go/no-go – Stroop – Backward digit span – Backward Corsi blocks – Visual 2-back). Both testing sessions were scheduled at the same time of the day, and lasted approximately one hour.

### Flanker

Participants viewed a series of arrows, either pointing to the left of the right of the screen. They were instructed to ignore all stimuli but the arrow at the center of the screen (target), and respond by pressing the left or right key when presented with arrows pointing left or right, respectively. For any given trial, the number of arrows displayed ranged from three to 25, with equal probability for congruent and incongruent trials. All sessions included 20 trials. We recorded accuracy and response time for both congruent and incongruent trials.

### Go/no-go

Participants were presented with a series of circles, either uniform or patterned. The uniform circle required a key response ('go') whereas the other required no response ('no-go'). If response was required, the stimulus remained visible indefinitely, until a response was made. When the stimulus required no response, it disappeared after 2000 ms. A self-paced button press triggered the start of the next trial. The interval from the button press to the presentation of the stimulus ranged from 500 ms to 2000 ms (randomly jittered). A session included between 12 and 40 trials.

### Stroop

Participants were presented with a series of color words, in a colored font either congruent or incongruent, drawn with equal probability. They were instructed to attend to the color of the font, and to respond by pressing the key corresponding to the appropriate color on the keyboard. Stimuli remained visible until a response was made. A session included between 20 and 50 trials. We recorded accuracy and response time for both congruent and incongruent trials.

### Backward digit span

Participants viewed a series of digits from 1 to 9 presented sequentially for 1000 ms, with 500 ms intertrial intervals. They were instructed to respond by entering the corresponding digits on the keyboard at the end of each trial. Hierarchical item randomization allowed the presentation of a maximum of two identical digits consecutively. For each trial, answers could be corrected until submitted. A self-paced button press triggered the start of the next trial. A session included 12 to 40 trials.

### Backward Corsi blocks

Participants were presented with a series of locations on a block, sequentially for 1000 ms, with 500 ms intertrial intervals. They were instructed to respond by clicking on the corresponding locations at the end of each trial. Hierarchical item randomization did not allow presentation of identical locations consecutively. For each trial, no correction was allowed once submitted. A self-paced button press triggered the start of the next trial. A session included 12 to 40 trials.

### Visual 2-back

Participants viewed a series of pictures presented sequentially for 2000 ms, with 500 ms intertrial intervals. They were instructed to press a key to signal a match, that is, two identical pictures interleaved with one stimulus in between (i.e., 2-back). No action was required in the absence of match. The number of matches ranged from 20 to 35 per session, randomized. A session included 40 to 70 trials.

## Physiological measurements

Physiological measures were collected using FitbitChargeHR$^{TM}$, powered by the MEMS tri-axial accelerometer. This multisensory wristband has shown adequate accuracy and reliability in previous studies for the measures of interest in the present study (e.g., *de Zambotti et al., 2016*). Measures included minutes of activity, calories burned, intensity, intensity range (sedentary, lightly active, fairly active, very active), steps and heart rate (measured by changes in blood volume using PurePulse$^{TM}$ LED lights).

## Questionnaire

Participants provided information about the following: ethnic background, age, gender, handedness, height, weight, diagnosis of learning disorder, brain trauma or epileptic seizures, current or past enrolment in a remediation or a cognitive training program, and whether English was their first language. In addition, self-reported information was gathered to quantify videogaming and physical exercise habits (4-point Likert scale in both cases), as well as to evaluate overall health, happiness, sleep quality, and mindset (6-point Likert scale for each item). The latter was intended to capture beliefs about the malleability of cognitive ability in the context of schoolwork, that is, the extent to which students perceive academic achievement in a predominantly fixed or malleable manner (see for example *Paunesku et al., 2015*). All measures were collected prior to the intervention, but variables susceptible to change over time were reassessed post-intervention.

## Genotyping

DNA collection was performed using Oragene-DNA Self-Collection kits, in a manner consistent with the manufacturer's instructions. DNA was subsequently extracted from all saliva samples according to a standardized procedure (*Nishita et al., 2009*). All resultant DNA samples were resuspended in Tris-EDTA buffer and were quantified used Nanodrop ND-1000 1-position spectrophotometer (Thermo Scientific, Waltham, MA, USA).

DNA samples were diluted to 50 ng/µL. A modified version of the method described by *Erickson et al., 2008* was used for DNA amplification. Amplification was carried out on the 113 bp polymorphic *BDNF* fragment, using the primers BDNF-F 5-GAG GCT TGC CAT CAT TGG CT-3 and BDNF-R 5-CGT GTA CAA GTC TGC GTC CT-3. Polymerase chain reaction (PCR) was conducted using 10X Taq buffer (2.5 L µL), Taq polymerase (0.125 µL), dNTPs (5 nmol), primers (10 pmol each), Q solution (5 µL), and DNA (100 ng) made up to 25 µL with dH2O. The PCR conditions consisted of

denaturation at 95°C for 15 min, 30 cycles on a ThermoCycler (involving denaturation at 94°C for 30 s, annealing at 60°C for 30 s, and extension at 72°C for 30 s) and a final extension at 72°C.

PCR product (6.5 µL) was incubated with Pm1l at 37°C overnight. The digestion products were analyzed using a high-resolution agarose gel (4%) with a Quick Load 100 bp ladder (BioLabs) and a GelPilot Loading Dye (QIAGEN). After immersion in an ethidium bromide solution for 10 min, DNA was visualized under ultraviolet light. Enzyme digestion resulted in a 113 bp fragment for the *BDNF* met[66] allele, and 78 and 35 bp fragments for the val[66] allele. This procedure is consistent with the one described by *Erickson et al. (2008)*.

## Intervention

Participants were randomly assigned to either an exercise group (*N* = 152) or a control (*N* = 153) group (see *Table 7*). Randomization was computer-based, generated in R (*Core Team R, 2016*) by one of the authors (D.M.). Group allocation was performed at the individual level. Testers were blind to group allocation.

The exercise intervention consisted of a high-intensity workout including the following: warm-up (2 min), short bursts (5 × 20 s, interleaved with incremental breaks (30 s, 40 s, 50 s, 60 s, and a shorter 20 s break after the last workout period), and stretching (2 min). The video-based workout did not require previous experience or knowledge, as it included basic fitness movements. All movements were designed so that participants could maintain their gaze fixed on the screen at all times. All instructions were provided both verbally (audio recording) and visually (on-screen captions). Complete details and script can be found in the online repository. A complete session lasted 10 min, and was scheduled every morning on weekdays. The control condition consisted of a blend of board games, computer games, and trivia quizzes, consistent with current recommendations regarding active control groups (*Boot et al., 2013*) and findings showing that aerobic exercise interventions typically do not differ from other regimens with respect to participants' expectations (*Stothart et al., 2014*). Consistent with this assumption, self-reported feedback indicated no difference in enjoyment or motivation between conditions, and no difference in mindsets regarding cognitive malleability (*Paunesku et al., 2015*).

Frequency and duration were matched between conditions. The intervention was 6 weeks long, with five sessions per week, for a total of 30 sessions. This translates to 300 min of actual exercise. There was no difference between groups regarding the number of sessions completed (*M* = 29.05, *SD* = 1.63, overall). Due to the nature of the intervention, class size was limited to 20 participants in both conditions. Participants were supervised at all times, to ensure a high degree of fidelity to the intended protocol. Participants did not differ between groups in any of the self-reported measures described previously, which include physical exercise habits. Note that participants did not exercise on the days of pretest and posttest, to prevent acute effects of physical exercise on cognitive performance (see *Tomporowski, 2003*).

**Table 7.** Demographics and sample characteristics at baseline.

|  | **HIT** | **Controls** | **Total** |
|---|---|---|---|
| *Sample (N)* | 152 | 153 | 305 |
| *Gender* | 90 f./62 m. | 97 f./56 m. | 187 f./118 m. |
| *Age* | 9.87 (1.81) | 9.96 (1.68) | 9.91 (1.74) |
| *Handedness (LH/Ambid.)* | 18/3 | 14/3 | 32/6 |
| *BMI* | 18.1 (3.92) | 18.51 (7.89) | 18.31 (6.25) |
| *LD diagnosis* | 13 | 16 | 29 |
| *Previous remediation* | 8 | 14 | 22 |
| *Videogaming* | 2.32 (0.95) | 2.43 (0.96) | 2.38 (0.97) |
| *Physical exercise* | 3.06 (0.8) | 2.95 (0.78) | 3.03 (0.81) |
| *Happiness* | 4.53 (1.25) | 4.61 (1.22) | 4.55 (1.27) |
| *Sleep quality* | 4.07 (1.36) | 4.11 (1.39) | 4.11 (1.41) |
| *General health* | 4.88 (1.05) | 4.84 (1.01) | 4.82 (1.06) |

## Acknowledgements

DM and KEW are supported by philanthropic donations from the Campus Link Foundation, the Kelliher Trust and Perpetual Guardian (as trustee of the Lady Alport Barker Trust). DM is also supported by the Neurological Foundation of New Zealand.

## Additional information

### Funding

| Funder | Grant reference number | Author |
|---|---|---|
| Centre for Brain Research | 9133-3706255 | David Moreau<br>Karen E Waldie |

The funders had no role in study design, data collection and interpretation, or the decision to submit the work for publication.

### Author contributions

David Moreau, Conceptualization, Data curation, Software, Formal analysis, Validation, Investigation, Visualization, Methodology, Writing—original draft, Project administration, Writing—review and editing; Ian J Kirk, Karen E Waldie, Supervision, Funding acquisition, Writing—review and editing

### Author ORCIDs

David Moreau http://orcid.org/0000-0002-1957-1941

### Ethics

Human subjects: We report here a multicenter, randomized (1:1 allocation), placebo-controlled trial. Design and reporting are consistent with CONSORT guidelines (http://www.consortstatement. org/). Participants, parents and school principals gave their informed consent for inclusion in this study, and the Ethics Committee at the University of Auckland approved all procedures (protocol #015078).

### Decision letter and Author response

Decision letter https://doi.org/10.7554/eLife.25062.sa1
Author response https://doi.org/10.7554/eLife.25062.sa2

## Additional files

### Supplementary files

- Reporting standard 1. CONSORT flow diagram.
- Reporting standard 2. CONSORT check list.

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
