## [Decision Letter]

Thank you for submitting your article "High-intensity Training Enhances Executive Function in Children" for consideration by *eLife*. Your article has been favorably evaluated by Richard Ivry (Senior Editor) and three reviewers, one of whom, Heidi Johansen-Berg (Reviewer #1), is a member of our Board of Reviewing Editors. The following individual involved in review of your submission has agreed to reveal their identity: Phillip Tomporowski (Reviewer #2).

The reviewers have discussed the reviews with one another and the Reviewing Editor has drafted this decision to help you prepare a revised submission.

Summary:

The reviewers found the topic interesting, found much value in the study and found the paper to be well written. They were impressed by the large sample and thorough statistical treatment of the data. However, some key methodological details were missing, making it impossible to judge the paper in its current form. Given the importance of the topic, the reviewers agreed to invite a revision to clarify these issues so that they could then take a decision on whether the study meets the standards required for publication in *eLife*.

Essential revisions:

1) The study design lacks details. Details are needed on how many schools were involved, how many children per school, how children/schools were randomised by condition and whether the statistical analyses take clustering effects (of school) into account.

2) The methods employed to measure children's cognitive performance lack sufficient detail. How were the tests administered; individually or group?

What was the testing environment?

Were the tests administered in the same order?

Could the authors comment on the role of proactive and retroactive interference?

There were a number of tests. What was the over-all duration of testing?

Could the authors comment on the possibility of changes in level of children's motivation? While an activity-control group was employed in this experiment, it clearly does not mean that the children are blinded to conditions, raising the possibility of expectancy effects altering children's motivation to perform during post testing. The authors attempt to address this issue (subsection “Intervention”). However, their assumptions concerning the motivational effects of exercise vs. non-exercise conditions were drawn from studies conducted with adults (Stothardt et al., 2014) or high school students (Paunesku, et al., 2015). There are clear developmental differences between the expectancies of 9-year old children's and those of adults.

Please justify the use of different number of trials based on individual asymptotes during each task, given that this approach is rarely used in the field.

Also, could the small number of trials (i.e. only 20 trials for flanker task, a minimum 12 trials for go/no go and working memory tasks) generate valid measure of cognitive control and working memory? Did each individual participant perform the same number of trials at pre- and post-test? If different numbers of trials were used, does such difference correlate with the performance change?

3) The exercise intervention lacks detail. Given the focus of the research is an implicit comparison between aerobic and HIT, it would seem important to detail the exercise regimen. Who instructed the children?

What was the average class size?

What instruction was provided?

What are "core" exercises?

Is practice required to perform these core exercises?

How were exercises timed?

What type of feedback was provided by instructors?

Were fidelity measures included in the training protocol? Without this information, it would be very difficult to replicate this experiment.

Also, the study used 80% of heart rate reserve (HRR) as the target HR during HIT workout. As far as we can tell, Figure 1C shows that the maximum HR during each workout appears to be approximately 140-150 in BPM, which is less likely to be equal to exercise intensity at 80% HRR. Instead, the HR data would suggest exercise intensity at light-to-moderate intensity in this age group according to the classification of physical activity intensity (ACSM, 2014). Please explain. Details of the HIT workout may also help explain why HR during the HIT workout did not meet the desired intensity.

4) As all data presented graphically are change scores it would be helpful to include summary scores on cognitive tests and fitness measures at for each timepoint so the reader can get a feel for how the children are performing on these tasks and what the differences translate to in terms of the raw scores.

5) Despite the significant findings, some readers may find it implausible that such a small amount of additional activity over a few weeks results in changes in cognitive performance. It would be helpful for the authors to say more about why this might be expected. Is there a threshold effect, for example, such that a small amount of additional activity may reach threshold for cognitive benefits?

It would also be helpful if they could provide data to support how much more daily/weekly energy expenditure the HIT intervention would deliver compared to the control activities. Such information regarding actual exercise volume/time (i.e. calories or 300 minutes) achieved over 6 weeks may be added to the limitation section, but meanwhile this information could also be used to suggest how much physical activity this intervention could contribute to the minimum daily/weekly recommendation for health benefit.

[Editors' note: further revisions were requested prior to acceptance, as described below.]

Thank you for resubmitting your work entitled "High-intensity Training Enhances Executive Function in Children" for further consideration at *eLife*. Your revised article has been favorably evaluated by a Senior Editor, a Reviewing Editor and three reviewers.

The manuscript has been improved but there are some remaining issues that need to be addressed before acceptance, as outlined below:

Essential revisions:

We are sorry for that it has taken some time to review the manuscript. In addition to consulting with the original reviewers we also sought input from a statistical reviewer to assess specifically the degree to which the study complies with the CONSORT guidelines for reporting of randomised controlled trials. Some changes are required to meet those guidelines:

In general, the reviewers agreed that transparency for this study is good, all the analysis scripts are made available through GitHUB. The randomisation procedure is clear and blinding has been considered.

Though the CONSORT checklist is reported, the paper has not been written in the spirit of the guidelines. This is a randomised trial where the effect of an intervention was compared to a control intervention for cognitive improvement in children. While the trial design is mentioned on the first page, CONSORT requires the trial be identified as randomised in the title. The Abstract should then focus on the design of the trial rather than be written in a narrative style.

Also the key result with point estimate and confidence interval needs to be presented in the Abstract. I realise that this is not the usual style for *eLife* but some concession should be made so that the CONSORT requirements are met.

CONSORT requires the primary and secondary outcomes to be specified. This report does not state which outcome is the primary outcome (usually this is the measurement that the sample size calculation was calculated for). The sample size statement in the protocol states 129 per group are required to detect an effect size of 0.35 with power of 80% at 5% significance level. This suggests that the study is powered for a single primary research question, but a primary outcome is not listed instead there are many outcomes listed under several sections. So it is not clear which outcome the difference of 0.35 relates to. While the statistical analysis proposed is Bayesian the arguments over repeated testing still hold if you want to present evidence that the intervention does improve outcomes. Results are frequently presented with statistical hypothesis tests and p-values, but the term 'statistically significant' is often abbreviated to 'significant' which is to be avoided. The authors need to be careful to always present the weight of the evidence in a p-value *and* also report a corresponding effect size with 95% confidence interval.

As a single primary outcome was not specified a priori it would be helpful if the Abstract communicated the size of the trial and the number of outcomes that were considered in the full protocol, so the statistical significance can be put in context.

---

## [Author Response]

Essential revisions:1) The study design lacks details. Details are needed on how many schools were involved, how many children per school, how children/schools were randomised by condition and whether the statistical analyses take clustering effects (of school) into account.

More information about the number of schools involved and characteristics of the participants have been added in the Participants section. Of particular relevance here:

“They were recruited from six schools across New Zealand, providing a sample of various socioeconomic deciles (three public institutions, three private), locations (three urban institutions, three rural), and ethnic backgrounds representative for the country (70% New Zealand European, 20% Pacific, 7% Asian, 3% Other). The number of students involved per school ranged from 5 to 83 (*M* = 50.8, *SD* = 31).”

Concerning randomization, we have added the sentence: “Group allocation was performed at the individual level”, to make it explicit that this was not a cluster randomized trial, but a trial in which every child was assigned to a group randomly.

More generally, details about the intervention have been added to the Materials and methods section and can be found with the source code we have provided (especially HIT_workout_script.txt). We discuss these thoroughly hereafter.

2) The methods employed to measure children's cognitive performance lack sufficient detail. How were the tests administered; individually or group?What was the testing environment?

This information has been added to the Cognitive Assessments section:

“Testing was conducted on school premises. All cognitive assessments were computer-based, administered in groups of a maximum of 15 students. This limit on the number of participants tested at a given time was implemented to minimize potential averse effects of group testing.”

Were the tests administered in the same order?

In the original manuscript, we stated: “The order below was the order of presentation for every participant at both pretest and posttest. Both testing sessions were scheduled at the same time of the day.” To ensure that this is explicit, we have added the order of presentation in parentheses. This passage now reads: “The order below was the order of presentation for every participant at both pretest and posttest (i.e., Flanker – Go/no-go – Stroop – Backward digit span – Backward Corsi blocks – Visual 2-back).”

Could the authors comment on the role of proactive and retroactive interference?

Because this comment comes immediately after the question about the order of tasks, we assumed that it was in relation to proactive and retroactive interference between tasks. The design was fixed with regard to tasks order, as it was more suitable to the hypotheses of the study. More specifically, we opted for a fixed order for the following reasons:

1) The main problem related to order effects is that it can confound treatment effects if the treatment and control conditions present problems in different orders. A fixed order solves most of the related problems;

2) Couterbalancing is important to establish the separate validity of each of the tasks, whereas we were interested in generating constructs to be compared across time, rather than comparing tasks with each other within a time point. In addition, counterbalancing is critical when a limited number of version for a given test exist a priori; in contrast, all our tasks when randomly generated;

3) Finally, with six tasks the number of permutations is 6! = 720. Even Latin Square designs fail to provide appropriate solutions, and introduce additional confounds.

For the reasons stated above, we opted for a fixed-order design, consistent with traditional methods in psychometric testing.

There were a number of tests. What was the over-all duration of testing?

We have added an estimate of duration for the cognitive testing session in the Materials and methods section: “Both testing sessions were scheduled at the same time of the day, and lasted approximately one hour.” As an aside, cognitive testing was reduced to the minimum possible without threatening tests validity (see our comment on the number of trials below).

Could the authors comment on the possibility of changes in level of children's motivation? While an activity-control group was employed in this experiment, it clearly does not mean that the children are blinded to conditions, raising the possibility of expectancy effects altering children's motivation to perform during post testing. The authors attempt to address this issue (subsection “Intervention”). However, their assumptions concerning the motivational effects of exercise vs. non-exercise conditions were drawn from studies conducted with adults (Stothardt et al., 2014) or high school students (Paunesku, et al., 2015). There are clear developmental differences between the expectancies of 9-year old children's and those of adults.

This is an excellent point, one that needs to be accounted for in any training design. We measured self-reported enjoyment and motivation after the intervention, and compared these between groups. This was initially reported in the Intervention section, but this is technically a finding that better belongs to the Results section, where it has now been edited:

“There was no difference between groups regarding self-reported enjoyment or motivation (*W* = 12058, *p* =.54 and *W* = 11497, *p* =.86, respectively). This finding allows controlling for expectation effects, and thus stronger causal claims (Boot, Simons, Stothart, and Stutts, 2013; Stothart, Simons, Boot, and Kramer, 2014)

This is in addition to the significant difference in terms of malleability *in favor of the Control group,* which suggests a potential for greater improvements in Controls:

“In addition, participants’ self-reported belief about cognitive malleability (i.e., mindset) indicated a significant difference (p <.03) in favor of the control group (*M* = 7.11, *SD* = 2.65 and *M* = 6.42, SD = 2.74, respectively).”

This further strengthens our main claim, namely that HIT elicits greater cognitive improvements than the control intervention, despite a more fixed mindset. We do not speculate on this effect in the main text of the paper, however, given that we did not hypothesize the effect a priori.

Please justify the use of different number of trials based on individual asymptotes during each task, given that this approach is rarely used in the field.Also, could the small number of trials (i.e. only 20 trials for flanker task, a minimum 12 trials for go/no go and working memory tasks) generate valid measure of cognitive control and working memory? Did each individual participant perform the same number of trials at pre- and post-test? If different numbers of trials were used, does such difference correlate with the performance change?

We made sure that reliability was adequate for the versions we proposed in the present experiment, via validation on an independent data set (Spearman-Brown prophecy formula). Trial length did not correlate with performance; information about reliability has been added to the Materials and methods section, to justify using the cognitive tasks in this manner:

“Unless specified otherwise, the number of trials varied based on individual performance to allow reaching asymptotes, with a minimum and a maximum specified for each task. […] Specifically, reliability was calculated by comparing test scores on the asymptotic version vs., the maximal-length version, for each task (see trial length details below and online repository for source code data).”

We corroborate this view with additional literature in the Discussion section:

“Despite time constraints, we aimed to preserve testing diversity (i.e., number of tasks per construct) for a few reasons. First, we strived to provide estimates of constructs that minimize task-specific components and extract meaningful domain-general scores. […] To further validate this approach, we piloted different versions of our testing tasks and determined that the validity of the versions we retained for the present study was acceptable, given appropriate reliability across task lengths.”

Finally, we included to the linked online repository the file Reliability_indices.csv, which contains all indices across tasks (subsection “Cognitive Assessments”).

3) The exercise intervention lacks detail. Given the focus of the research is an implicit comparison between aerobic and HIT, it would seem important to detail the exercise regimen. Who instructed the children?

This information has been added to the Intervention section:

“All instructions were provided both verbally (audio recording) and visually (on-screen captions). Complete details and script can be found in the online repository.”

What was the average class size?

This information has been added to the Intervention section:

“Due to the nature of the intervention, class size was limited to 20 participants in both conditions.”

What instruction was provided?

The complete script of the HIT workout is freely available online (https://github.com/davidmoreau/HIT). The file HIT_workout_script.txt includes all instructions and triggers of the video.

What are "core" exercises?

In the initial version of the manuscript, we used the terminology “core” of the session to describe the high-intensity part of the training. For clarity, this term has now been replaced by the more explicit “short bursts”, which we believe describes more accurately this part of the protocol.

Is practice required to perform these core exercises?

This is an important aspect of the workout that was not emphasized enough in the initial version of the manuscript. We have now added:

“The video-based workout did not require previous experience or knowledge, as it included basic fitness movements. […] All instructions were provided both verbally and on-screen. Complete details and script can be found in the online repository.”

For the interested reader, and for the purpose of replicability, all instructions and second-by-second details of the workout can be found in the online repository, as mentioned:

“R code and data are freely available on GitHub (https://github.com/davidmoreau). The repository includes data sets, R scripts, details and script of the HIT workout, the CONSORT flow diagram and the CONSORT checklist.”

How were exercises timed?

Timing was maintained via screen-based instructions, and ensured by adult supervision. We made this explicit in the manuscript with the following: “Participants were supervised at all times, to ensure a high degree of fidelity to the intended protocol”. Adequate fidelity to the protocol was also ensured post-session with the minutes of activity measure provided by activity trackers.

What type of feedback was provided by instructors?

No feedback was provided by the instructors, to avoid introducing confounds in the intervention. This was meant to allow more rigorous matching between conditions (HIT vs. control).

Were fidelity measures included in the training protocol? Without this information, it would be very difficult to replicate this experiment.

Fidelity to the intended protocol was fundamental to the present experiment, given that the rationale is about intensity of physical exercise. We reiterate this point in the Discussion:

“This indicates a high degree of agreement, or fidelity, to the intended protocol—an essential component of the intervention given the underlying assumption that participants would exercise at a high intensity. […] Arguably, this adaptive property emerged from the design of the intervention, whereby participants were encouraged to exercise at maximum intensity at the time of the workout—an intrinsically individual and dynamic variable by definition.”

More specifically, this was the rationale for monitoring exercise via activity trackers, and for presenting the accuracy ratio, a measure of fidelity to the protocol (see figure below). To make sure that this is further emphasized in the paper, we have added the following to the Discussion:

“We could thus ensure that training was adequate, performed at a suitable intensity, and could test directly the dynamic coupling of these variables and their effects on cognitive outcomes. Indeed, we found that almost all participants stayed within the overall targeted range of effort required by the design of the workout and by initial individual measurements.”

We further detail the importance of controlling for the intensity of exercise in this type of design:

“This result emphasizes the importance of controlling exercise intensity in HIT studies, given that the main determinant of improvement appears to be the intensity of the workout. […] The high-intensity component of this type of exercise regimen is intended to allow for higher workout intensity than traditional workouts, despite shorter overall volume.”

Also, the study used 80% of heart rate reserve (HRR) as the target HR during HIT workout. As far as we can tell, Figure 1C shows that the maximum HR during each workout appears to be approximately 140-150 in BPM, which is less likely to be equal to exercise intensity at 80% HRR. Instead, the HR data would suggest exercise intensity at light-to-moderate intensity in this age group according to the classification of physical activity intensity (ACSM, 2014). Please explain. Details of the HIT workout may also help explain why HR during the HIT workout did not meet the desired intensity.

In the initial version of the manuscript, Figure 1C showed*δ* HR_Max_ averaged over all participants for a given session, as a function of session. This was thought to better convey both HR_Max_ and *δ* in a single graph (HR_Target_ = HR_Rest_ + *δ* HR_Max_ – *δ* HR_Rest_). However, we acknowledge that this may be confusing for the reader, for the reasons outlined by the reviewers. Therefore, we have now rescaled the ordinate to show uncorrected HR_Max_ (i.e. measured HR_Max_) values across sessions. Combined with Figure 1D, Figure 1C shows the adaptive property of the workout – namely, maximum heart rate is maintained over time, despite decreasing resting heart rate, to accommodate increasing workout volume. The complete script of the HIT workout is freely available online (https://github.com/davidmoreau/HIT). The file HIT_workout_script.txt includes all instructions and triggers of the video.

4) As all data presented graphically are change scores it would be helpful to include summary scores on cognitive tests and fitness measures at for each timepoint so the reader can get a feel for how the children are performing on these tasks and what the differences translate to in terms of the raw scores.

This is an excellent point—we have added a new table that presents descriptive for each task and measurement (Table 6).

5) Despite the significant findings, some readers may find it implausible that such a small amount of additional activity over a few weeks results in changes in cognitive performance. It would be helpful for the authors to say more about why this might be expected. Is there a threshold effect, for example, such that a small amount of additional activity may reach threshold for cognitive benefits?

We appreciate the opportunity to clarify our contribution. We have further described the hypothesized mechanisms of improvements in the Introduction and the Discussion.

The Introduction now states:

“In some cases, physiological improvements following high-intensity training (HIT) can even go beyond those typically following aerobic regimens (Rognmo, Hetland, Helgerud, Hoff, and Siørdahl, 2004). […] Together, the conjunction of promising findings and clear mechanisms of action has prompted discussions to implement HIT interventions more systematically within the community (Gray, Ferguson, Birch, Forrest, and Gill, 2016).”

The Discussion details further the postulated mechanisms of improvement:

“Regardless of the strength of the evidence reported in the present study, one might question the genuine impact of such a short training regimen. […] In terms of practical implications, this aspect is critical, as it allows designing shorter, more potent workouts.”

It would also be helpful if they could provide data to support how much more daily/weekly energy expenditure the HIT intervention would deliver compared to the control activities. Such information regarding actual exercise volume/time (i.e. calories or 300 minutes) achieved over 6 weeks may be added to the limitation section, but meanwhile this information could also be used to suggest how much physical activity this intervention could contribute to the minimum daily/weekly recommendation for health benefit.

We have added this information to the Materials and methods section:

“Frequency and duration were matched between conditions. The intervention was 6-weeks long, with five sessions per week, for a total of 30 sessions. This translates to 300 min of actual exercise.”

The Discussion section provides more details about the cardio-metabolic outcomes of HIT interventions, and therefore puts in context the overall low volume of exercise in the intervention:

“In a review of the impact of HIT on public health, Biddle and Batterham (2015) go as far as to premise their entire argument on the idea that the cardio-metabolic health outcomes are not to be questioned – rather, their only concern was about whether or not such regimens can be widely implemented and sustained over time (Biddle & Batterham, 2015). In a similar vein, Costigan and colleagues concluded that HIT is a time-efficient approach for improving cardiorespiratory fitness in adolescent populations (Costigan, Eather, Plotnikoff, Taaffe, and Lubans, 2015).

However, it is important to note that exercise recommendations are often expressed in terms of time commitment rather than workout intensities, and that these figures are typically framed in the context of aerobic exercise activities (e.g., Manore et al., 2014). Therefore, estimates based on these figures and applied to non-aerobic interventions may be fundamentally inaccurate.

[Editors' note: further revisions were requested prior to acceptance, as described below.]

Essential revisions:We are sorry for that it has taken some time to review the manuscript. In addition to consulting with the original reviewers we also sought input from a statistical reviewer to assess specifically the degree to which the study complies with the CONSORT guidelines for reporting of randomised controlled trials. Some changes are required to meet those guidelines:In general, the reviewers agreed that transparency for this study is good, all the analysis scripts are made available through GitHUB. The randomisation procedure is clear and blinding has been considered.Though the CONSORT checklist is reported, the paper has not been written in the spirit of the guidelines. This is a randomised trial where the effect of an intervention was compared to a control intervention for cognitive improvement in children. While the trial design is mentioned on the first page, CONSORT requires the trial be identified as randomised in the title.

The title has been edited to:

“High-intensity Training Enhances Executive Function in Children in a Randomized, Placebo-Controlled Trial”.

The Abstract should then focus on the design of the trial rather than be written in a narrative style.Also the key result with point estimate and confidence interval needs to be presented in the Abstract. I realise that this is not the usual style for eLife but some concession should be made so that the CONSORT requirements are met.

The Abstract has been edited to meet the CONSORT requirements, while ensuring it remains within the 150-word limit. In particular, primary and secondary outcomes are now explicit:

“318 children aged 7-13 years were randomly assigned to a HIT or an active control group matched for enjoyment and motivation. In the primary analysis, we compared improvements on six cognitive tasks representing two cognitive constructs (*N* = 305). Secondary outcomes included genetic data and physiological measurements.”

In addition, the main result is now presented in the Abstract, in the “Results” subsection:

“The 6-week HIT regimen resulted in improvements on measures of cognitive control [BF_M_ = 3.38, g = 0.31 (0.09, 0.54)] and working memory [BF_M_ = 5233.68, *g* = 0.54 (0.31, 0.77)], moderated by BDNF genotype, with met^66^carriers showing larger gains post-exercise than val^66^ homozygotes.”

CONSORT requires the primary and secondary outcomes to be specified. This report does not state which outcome is the primary outcome (usually this is the measurement that the sample size calculation was calculated for). The sample size statement in the protocol states 129 per group are required to detect an effect size of 0.35 with power of 80% at 5% significance level. This suggests that the study is powered for a single primary research question, but a primary outcome is not listed instead there are many outcomes listed under several sections. So it is not clear which outcome the difference of 0.35 relates to. While the statistical analysis proposed is Bayesian the arguments over repeated testing still hold if you want to present evidence that the intervention does improve outcomes.

Thank you for pointing this out. The primary and secondary outcomes are now clearly labeled in the Abstract and in text (subsection “Cognitive Improvements”). In addition, the power analysis now explicitly refers to the primary outcome measures:

“Note that an a priori power analysis based on previous studies (Erickson et al., 2013; Moreau et al., 2015) indicated the need for a minimum *N* of 129 participants per group to detect an effect of *d* = 0.35 on the primary outcome measures, with 1 – *β* =.80 and *α* =.05.” This allowed us to define our sample size based on the effects observed in Erickson et al., 2013 and Moreau et al., 2015. In addition to focusing on well-defined constructs, rather than multiple single measures, we also used Bayesian methods (both hypothesis testing and parameter estimation) that are robust to typical problems associated with multiple comparisons (Dienes, 2008; Gelman, 2008). Of course, this argument falls short when presenting the frequentist analyses, but those are reported for transparency and to reach a wider audience, and therefore are “equivalent” in the sense of being the typical alternative tests, yet not necessarily in the exact interpretation of the results. With this in mind, the interaction effects for Cognitive Control and Working Memory (*p* =.008 and *p* <.001, respectively) still remain largely significant after a Bonferroni correction.

Results are frequently presented with statistical hypothesis tests and p-values, but the term 'statistically significant' is often abbreviated to 'significant' which is to be avoided. The authors need to be careful to always present the weight of the evidence in a p-value and also report a corresponding effect size with 95% confidence interval.

Thank you for pointing out this oversight. The term ‘significant’, when referring to statistical significance, has now been systematically replaced with the latter (see “Frequentist Analyses” section). In addition, all effect sizes (*g, η*^2^) are now accompanied by 95% confidence intervals (we have replaced Cohen’s *d* with Hedges’ *g*, to correct for bias when sample sizes are unequal between groups). RMSE have also been added to all *R*^2^ values. Finally, the error term in Bayesian analyses, expressed in percentage, is also a measure of uncertainty (about the model, in this case). These are included in all tables for model comparison (2, 3, and 4).

As a single primary outcome was not specified a priori it would be helpful if the Abstract communicated the size of the trial and the number of outcomes that were considered in the full protocol, so the statistical significance can be put in context.

This information has been added to the Abstract in its new format:

“318 children aged 7-13 years were randomly assigned to a HIT or an active control group matched for enjoyment and motivation. In the primary analysis, we compared improvements on six cognitive tasks representing two cognitive constructs (*N* = 305). Secondary outcomes included genetic data and physiological measurements.”

Word space (150) limits our ability to provide details in the Abstract, but all the outcomes measured are listed in the main manuscript (Materials and methods) and in the Statistical Analysis Plan.